# Discovery of novel ancestry specific genes for androgens and hypogonadism in Million Veteran Program Men

Meghana S. Pagadala [1,2,3], Craig C. Teerlink[4,5], Guneet K. Jasuja[6,7,8], Madhuri Palnati[6], Tori Anglin-Foote [4], Nai-Chung N. Chang [4], Rishi Deka[1,9,10], Kyung M. Lee [4], Fatai Y. Agiri[4], Tiffany Amariuta[11,12], Tyler M. Seibert [1,9,13,14], Brent S. Rose[1,9,15], Kathryn M. Pridgen[4], Julie A. Lynch [4,5], Hannah K. Carter [11], Matthew S. Panizzon[1,10,16,18] & Richard L. Hauger [10,16,17,18] ✉

Given the various roles of testosterone in men's health, we conducted a multi-ancestral genetic analysis of total testosterone, free testosterone, SHBG, and hypogonadism in men within the Million Veteran Program (MVP). Here we identified 157 significant testosterone genetic variants, of which 8 have significant ancestry-specific associations. These variants implicate several genes, including *SERPINF2*, *PRPF8*, *BAIAP2L1*, *SHBG*, *PRMT6*, and *PPIF*, related to liver function. Genetic regulators of testosterone have cell type-specific effects in the testes, liver, and adrenal gland and are associated with disease risk. We conducted a meta-analysis amongst ancestry groups to identify 188 variants significantly associated with testosterone, of which 22 are novel associations. We constructed genetic scores for total testosterone, SHBG levels, and hypogonadism and find that men with higher testosterone genetic scores have lower odds of diabetes, hyperlipidemia, gout, and cardiac disorders. These findings provide insight into androgen regulation and identify novel variants for disease risk stratification.

Testosterone is an anabolic steroid synthesized by the testes which acts as the primary sex hormone in men[1–3]. Classical, well-established roles of testosterone include regulating sexual development during puberty, spermatogenesis, erythropoiesis, muscular strength, and bone density mass[4,5]; however, testicular androgens also have extensive physiological effects on cardiovascular, metabolic, hepatic, immune, and brain function[2,6–8]. These effects are primarily mediated by the androgen receptor (AR), which is encoded on the X

[1]Research Service, VA San Diego Healthcare System, San Diego, CA, USA. [2]Medical Scientist Training Program, University of California San Diego, La Jolla, CA, USA. [3]Biomedical Science Program, University of California San Diego, La Jolla, CA, USA. [4]VA Informatics and Computing Infrastructure (VINCI), VA Salt Lake City Healthcare System, Salt Lake City, UT, US. [5]Department of Internal Medicine, Division of Epidemiology, University of Utah School of Medicine, Salt Lake City, UT, US. [6]Center for Healthcare Organization and Implementation Research (CHOIR), VA Bedford Healthcare System, Bedford, MA, US. [7]Section of General Internal Medicine, Boston University School of Medicine, Boston, MA, US. [8]Department of Health Law, Policy, and Management, Boston University School of Public Health, Boston, MA, US. [9]Department of Radiation Medicine and Applied Sciences, University of California San Diego, La Jolla, CA, USA. [10]Department of Psychiatry, University of California San Diego, La Jolla, CA, USA. [11]Department of Medicine, University of California San Diego, La Jolla, CA, USA. [12]Halicioglu Data Science Institute, University of California San Diego, La Jolla, CA, USA. [13]Department of Radiology, University of California San Diego, La Jolla, CA, USA. [14]Department of Bioengineering, University of California San Diego, La Jolla, CA, USA. [15]Department of Urology, University of California San Diego, La Jolla, CA, USA. [16]Center for Behavior Genetics of Aging, University of California San Diego, La Jolla, CA, USA. [17]Center of Excellence for Stress and Mental Health (CESAMH), VA San Diego Healthcare System, San Diego, CA, USA. [18]These authors contributed equally: Matthew S. Panizzon, Richard L. Hauger. ✉e-mail: rhauger@health.ucsd.edu

chromosome and is widely expressed in many tissues throughout the body, particularly in the testes, prostate, liver, adipose tissue, skeletal muscle, heart, kidney, and brain[2,9]. Progressive reduction in testicular androgen steroidogenesis begins in the third to fourth decade of a man's life, with a roughly 1–3% decrease in circulating levels of total testosterone per year[1,2,10,11]. The development of age-related testosterone deficiency or hypogonadism contributes to sexual dysfunction, frailty, sarcopenia, fatty liver disease, cardiovascular disorders, chronic kidney disease, obesity, and metabolic dysregulation[1,2,10–19]. Testosterone deficiency has also been linked to depression, neurodegeneration, cognitive impairment, and Alzheimer's disease risk during male aging[20–25].

Given the various roles of testosterone in men's health, ascertaining genetic determinants of testosterone is vital. Approximately 50–60% of the testosterone in circulation is bound to sex hormone-binding globulin (SHBG), and 40–50% is bound to albumin[10]. Unbound testosterone (1–2%), which is termed free testosterone, and SHBG also function as important modulators of androgen action. To date, genome-wide association studies (GWAS) of testosterone, SHBG, and low testosterone levels have unveiled associations with variants in or near genes such as *SHBG, JMJD1C, FKBP4, REEP3*, and *FAM9B* and suggest that the SNP-based heritability for total and free testosterone is ~ 20% and ~ 15%, respectively[26–34]. These GWAS studies have been conducted primarily in European or Asian ancestry groups, despite the differences in testosterone levels that have been observed in men of different ancestries[35–37]. Few studies have analyzed ancestry-specific and trans-ancestry genes regulating total testosterone, free testosterone, SHBG, and hypogonadism in a diverse male cohort.

In this study, we present a GWAS of total and free testosterone levels, SHBG levels, and hypogonadism in the Million Veteran Program (MVP), a large, multi-ethnic genetic biorepository from a hospital-based population. As a hospital-based population, MVP has a higher disease prevalence of testicular dysfunction/hypofunction, type 2 diabetes, obesity, hyperlipidemia, prostate cancer, and cardiovascular diseases compared to other population studies, thus offering unique insights into the interaction of diseases and testosterone-regulating genes in a clinical setting. We identify associations with total testosterone, free testosterone, SHBG levels, and hypogonadism risk within European (EUR), African (AFR), admixed American (AMR), and East Asian (EAS) ancestry groups. In colocalization analyses, we find that several variants associated with testosterone levels and hypogonadism risk are located in or near genes related to liver function (*UGT2B17, BRI3, PRMT6, PPIF)*. Lastly, we construct and validate genetic scores and genetic risk scores (GS/GRSs) to estimate testosterone, SHBG levels, and hypogonadism and find significant associations with metabolic conditions, such as hyperlipidemia, gout, and diabetes, as well as cardiac and liver disorders.

## Results
### Curation of Laboratory Measures and Identification of Hypogonadism
We identified male participants within the MVP with documented total testosterone levels ($n = 265,038$), free testosterone levels ($n = 110,627$), and SHBG levels ($n = 43,325$). After initial data curation and pre-processing, morning total testosterone, free testosterone, and SHBG levels were available for 139,226, 49,709, and 19,635 men, respectively (Supplementary Fig. S1 and Table 1). MVP phenotyping data is dependent on endocrine lab tests being ordered in a clinical setting, resulting in different sample sizes for each hormone level. Average total testosterone (avg +/− std dev: 389 [196–582] ng/dL), free testosterone (9.4 [− 26–45] ng/dL), and SHBG (42 [15–68] nmol/L) levels were within the physiologically normal range for men (Supplementary Fig. S2A–C). Notably, our analysis confirmed the well-documented trend of decreasing total and free testosterone levels and increasing SHBG levels with age (Supplementary Fig. S2D–F)[38,39].

MVP ancestry groups were assigned using a recently published random-forest clustering with the 1000 Genomes Project and the Human Genome Diversity Project (1kGP + HGDP) reference panels[40]. Differences in total testosterone, free testosterone, and SHBG levels based on ancestry were small (Supplementary Fig. S2G–I). Significantly higher total testosterone levels were observed within the AFR ancestry group (406.6 [200.5–612.7] ng/dL) compared to EUR (382.9 [193.5–572.4] ng/dL), EAS (382.8 [201.8–563.7] ng/dL), and AMR (395.1 [201.9–588.4] ng/dL) ancestry groups ($p < 0.001$) (Supplementary Fig. S2G). In addition, the EAS ancestry group had lower SHBG levels (30.8 [12.4–49.3] nmol/L) compared to other ancestry groups (EUR: 42.7 [16.9–68.6] nmol/L, AFR: 41.2 [11.3–71.2] nmol/L, AMR: 36.4 [11.9–60.9] nmol/L; $p < 0.001$) (Supplementary Fig. S2I).

We extended our analysis to hypogonadism, a condition characterized by abnormally low testosterone levels. Based on our criteria for hypogonadism diagnosis (Methods), ~ 25% of the study population have hypogonadism; the incidence was higher in the AFR and AMR ancestry groups and lower in the EAS ancestry group (Supplementary Fig. S2J). We confirmed that men with hypogonadism had lower total testosterone levels compared to matched controls (Supplementary Fig. S2K).

Before conducting GWAS, we estimated heritability or the proportion of phenotypic variance due to genetic variance for testosterone phenotypes using linkage disequilibrium score regression (LDSC) analysis[41]. Total testosterone, SHBG, and hypogonadism heritability estimates were 9–12%, 20–50%, and 5–9%, respectively (Supplementary Table S1). The positive genetic correlation was observed for total testosterone with free testosterone and SHBG levels in the EUR ancestry groups, while negative genetic correlations were observed between total testosterone and hypogonadism in the EUR, AFR, and AMR ancestry groups (Supplementary Table S2).

### Genetic associations of laboratory measures and hypogonadism
We conducted GWAS with total testosterone levels, free testosterone levels, SHBG levels, and hypogonadism within MVP ancestry groups[40]. We identified men of EUR (95,184), AFR (27,521), AMR (13,443), and EAS (1816) ancestry for total testosterone analysis. Cohort sizes for free testosterone, SHBG, and hypogonadism are given in Table 1.

We discovered 157 significant genetic associations for total and free testosterone, SHBG, and hypogonadism (Supplementary Fig. S3 and Supplementary Data 1). Of these, the most significant associations were from analyses of EUR (98) and AFR (35) ancestry groups; however, significant associations in AMR (16) and EAS (8) groups were also identified. Eight variants had significantly different effect size estimates when comparing ancestry groups ("Methods"); most variants had different effect sizes in the EAS (5) ancestry group compared to EUR (0), AFR (1), and AMR (2) ancestry groups (Supplementary Fig. S4). Generally, however, variants had similar effects on hormone levels across ancestry groups.

**Table 1 | Sample Sizes for GWAS Analysis of Testosterone Levels and Hypogonadism**

| Ancestry Groups | Genotype | Total | Free | SHBG | Hypogonadism |
|---|---|---|---|---|---|
| All | 588177 | 139226 | 49709 | 19635 | 146720 |
| European | 421613 | 95184 | 33470 | 13893 | 102635 |
| African | 105305 | 27521 | 9863 | 2960 | 27736 |
| Admixed American | 50829 | 13443 | 5256 | 2081 | 13328 |
| East Asian | 9154 | 1836 | 545 | 471 | 1690 |

Table of sample sizes for all EUR, AFR, AMR, and EAS MVP men used for GWAS of total testosterone (total), free testosterone (free), SHBG, and diagnosis of hypogonadism. Also, included number of individuals for which genotypes were available.

We performed a SuSiE fine-mapping analysis of 157 associations to determine the proportion of variants with a high probability of being causal. An important advantage of SuSiE is that this analysis method allows for multiple causal variants in a region, in contrast to other causality inference methods[42]. The associations of 49 (31.2%) lead variants had a high PIP > 0.10. Several of these variants were missense variants, implicating genes such as *SERPINA1, NR2F6, FKBP4, SLCO1B1, HGFAC*, and *UGT2B15*.

To replicate our results, we compared significant associations from MVP EUR GWAS with total testosterone, free testosterone, and SHBG associations from the UK Biobank analyses performed by Ruth et al. with total testosterone, free testosterone, and SHBG (hypogonadism GWAS was not conducted)[26]. Of the MVP EUR GWAS variants, 87.2% (89.1% total testosterone, 100% SHBG, 33.3% free testosterone) were replicated in the UK Biobank. Six MVP EUR variants were not replicated, but effect sizes and minor allele frequencies were generally consistent (Supplementary Data 2 and Supplementary Fig. S5). We also compared MVP AFR GWAS with associations with total testosterone, free testosterone, and SHBG levels for 1,257 African-American men in the Multi-Ethnic Study of Atherosclerosis (MESA) cohort. Only two variants (14%) were nominally significant ($p < 0.05$) in the MESA cohort, likely due to its smaller sample size. However, effect size estimates of associations were consistent with MVP associations (Pearson $r = 0.64$, $p < 0.005$) (Supplementary Fig. S6).

### Gene expression in specific cell types and disease risk

To identify candidate genes influencing testosterone levels, we performed colocalization analysis using GTEx cis-eQTL gene expression summary statistics for androgen-regulated tissues and/or sites and MVP EUR GWAS. The colocalization analysis estimates the probability of a shared causal signal between *cis*-eQTL and GWAS analyses of total testosterone levels, free testosterone levels, SHBG levels, and hypogonadism[43]. In total, we discovered 24 genes with high probability (PP.H4 > 0.8) of shared causal variants between their expression and total testosterone levels (15), hypogonadism (17) and SHBG levels (5) (Fig. 1). Genes such as *SHBG, SERPINF2, PRPF8, PRMT6, BRI3, BAIAP2L1, NF1, BMP8A, PABPC4, TUFM, PPIF, SULT1A1, NYNRIN* and *RP11-327J17.2* have significant shared colocalization signal with total testosterone levels, SHBG levels, or hypogonadism (Supplementary Data 3). *NYNRIN, TUFM*, and *SH2B1* were uniquely identified by colocalization analysis of hypogonadism, while *NF1, PABPC4*, and *SHROOM3* were uniquely identified through colocalization analysis of total testosterone. We found that some gene associations (e.g., *SULT1A1, PRPF8, and SERPINF2*) were specific to the testes while *BRI3* and *NYNRIN* had liver cell type-specific effects. Although the testes are the primary source of testosterone production, these results identify the potential involvement of other tissues and organs for testosterone regulation.

To evaluate the disease relevance of the testosterone and hypogonadism variants, we performed PheWAS analyses with 1875 phecodes in the MVP. Testosterone variants were associated with diabetes ($n = 8$), hyperlipidemia ($n = 8$), gout ($n = 6$), and liver disease ($n = 7$). To determine whether PheWAS associations were due to testosterone changes, we ran colocalization analyses to determine the probability of shared causal variants between testosterone and disease. Approximately 85.9% (367/427) of the PheWAS associations had a greater than 80% probability of shared causal variants (Supplementary Data 4).

### MVP Meta-analysis of laboratory measures and hypogonadism

After the ancestry GWAS was completed, we conducted a meta-analysis to identify a consensus set of 188 associations affecting total testosterone, free testosterone, SHBG, and hypogonadism (Fig. 2 and Supplementary Data 5). Most variants were associated with total testosterone and hypogonadism, as these analyses had a larger sample size and thus were more powered to identify significant associations. Generally, effect sizes were consistent across ancestry groups

(Supplementary Fig. S7). We identified several genes implicated in previous testosterone GWAS analyses, such as *JMJDC1, SERPINF2, BAIAP2L1, FKBP4*, and *SHBG* (Fig. 3)[26–34]. Most variants were non-coding; 14 associations were missense variants, resulting in protein changes. Several missense variants were in liver function genes, such as *SERPINA1, NR2F6, GCKR, SLCO1B1, PNPLA3, HGFAC* and *UGT2B15*. In fact, 31% (53/171) of lead variants were significantly associated with AST and ALT levels, underscoring the effects of variants on liver function.

We compared our variants to previous analyses of total testosterone, free testosterone, and SHBG levels conducted in the UK Biobank[44]. Notably, all participants in the UK Biobank have total testosterone, SHBG, and bioavailable testosterone measured upon enrollment, whereas phenotype measurements were only available in MVP if ordered by a physician. The MVP cohort has a higher disease prevalence of testicular dysfunction/hypofunction, type 2 diabetes, obesity, hyperlipidemia, prostate cancer, gout, and sleep apnea compared to the UK Biobank (Table 2). 64.1% (75/117) of variants were significant in UK Biobank analyses. Since a majority of variants were found on chromosome X, we removed X chromosome variants in LD ($R^2 > 0.2$) with other X chromosome novel variants, leaving 22 novel MVP variants/loci (Supplementary Data 6).

Hypogonadism summary statistics for the UK Biobank were not available. However, only 416 men had a diagnosis code of testicular hypofunction (International Classification of Disease [ICD-10] E29.1, 0.09%) in the UK Biobank in contrast to the 25% of men within the MVP cohort who meet this study's definition of hypogonadism; thus, a case-control GWAS in UK Biobank would be severely underpowered.

### GS and GRS Construction and validation

Given the number of disease associations with identified testosterone variants, we constructed GS/GRS including significant meta-analysis associations for each phenotype. The GRS was used to evaluate hypogonadism risk, while the GS evaluated lab values rather than risk. The GS effectively stratified MVP men by levels of total testosterone and SHBG levels (Fig. 4A, B). Free testosterone GS was not evaluated because < 5 significant variants were identified. The odds of hypogonadism were evaluated for each decile of hypogonadism GRS (Fig. 4C). The odds of hypogonadism in top deciles for European (EUR) (1.10 [1.10–1.11]), African (AFR) (1.06 [1.05–1.08]) and American (AMR) (1.09 [1.07–1.11]) ancestry groups were significantly greater than those in bottom deciles for each ancestry group. We validated our total testosterone and SHBG GRS in the UK Biobank (Supplementary Fig. S8A, S8B). The hypogonadism GRS did not appear to stratify odds in the UK Biobank (Supplementary Fig. S8C).

Top significant associations from PheWAS analyses included metabolic (Phecode 250-Type 2 Diabetes, Phecode 272-Hyperlipidemia, Phecode 274-Gout, Phecode 278-Obesity), liver (Phecode 571, 572, 573) and cardiac disorders (Phecode 401, 411, 427). To determine if total testosterone and SHBG levels were associated with disease risk, we first conducted Mendelian randomization[45,46] studies with MVP EUR results and published GWAS studies of gout, type 2 diabetes, obesity, metabolic dysfunction-associated steatotic liver disease (MASLD), hyperlipidemia, and chronic heart failure (Supplementary Data 7). Lower levels of total testosterone and SHBG were associated with a higher risk of hyperlipidemia, while hypogonadism was associated with a higher risk of hyperlipidemia. Hypogonadism was causally associated with a higher risk of gout (effect size = 0.40, $p < 0.03$), while higher levels of SHBG was associated with a lower risk of chronic heart failure (effect size = −0.09, $p < .04$) (Supplementary Fig. S9).

To determine whether our constructed and validated GS could reveal disease associations in MVP ancestry groups, we conducted Cox proportional hazards analyses with the total testosterone GS, the SHBG GS, and the hypogonadism GRS (Fig. 5 and Supplementary Data 8). Higher total testosterone levels were associated with lower risk of type 2 diabetes (0.97 [0.96-0.98], $p < 10^{-4}$) and cardiac disorders

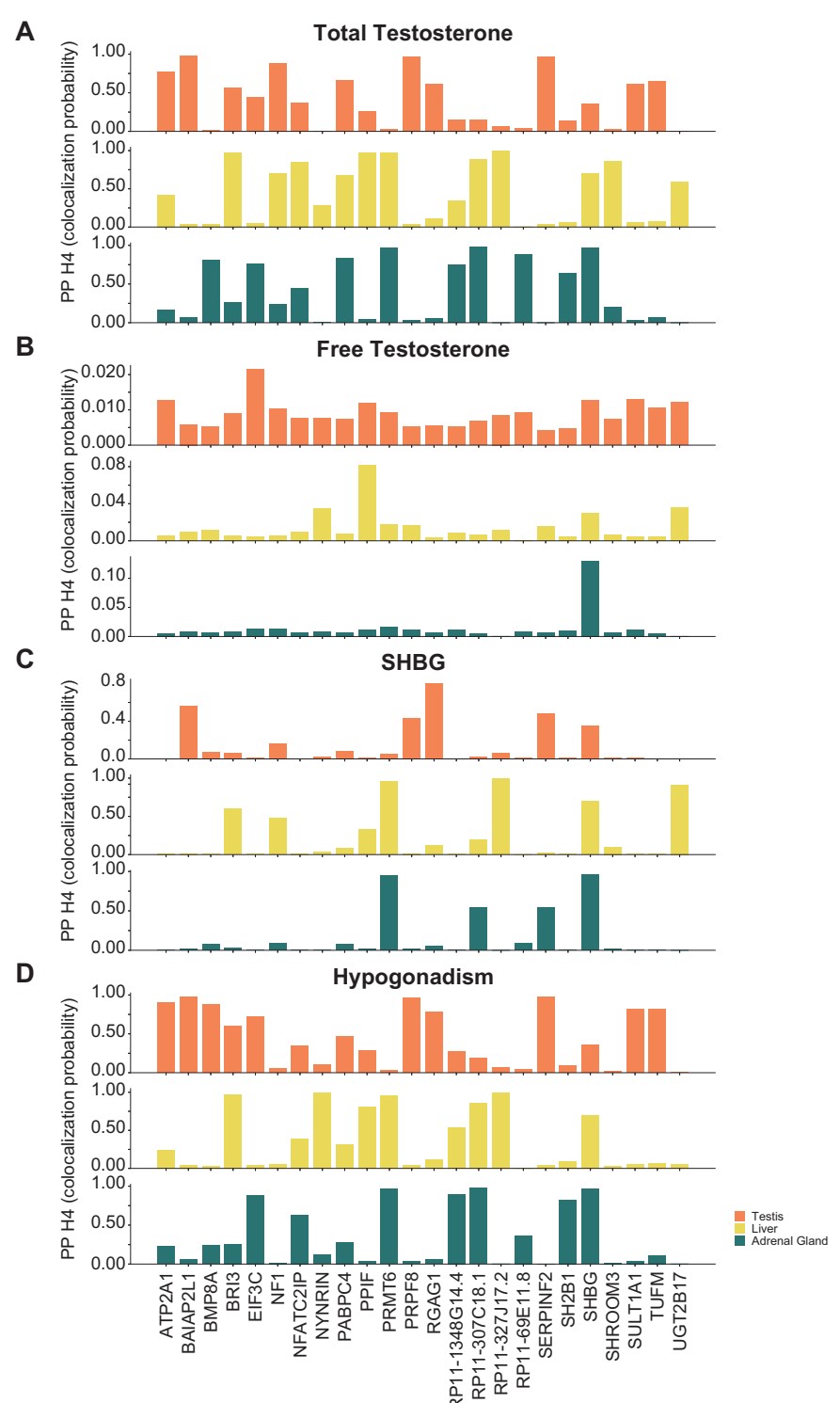

**Fig. 1 | Testosterone variants affect gene expression in specific cell types.** Bar plot of colocalization probability (PP.H4) with GTEx gene expression in testes (orange), liver (yellow), adrenal gland (green) for (**A**) total testosterone, (**B**) free testosterone, (**C**) sex-hormone binding globulin (SHBG) levels and (**D**) hypogonadism. Full colocalization results are given in Supplementary Data 3.

(0.97 [0.96-0.97], $p < 10^{-9}$) in the EUR ancestry group (Fig. 5A, F). Higher SHBG levels were associated with a lower risk of gout (0.92 [0.90–0.93], $p < 10^{-6}$) in the EUR ancestry group (Fig. 5D). Risk of hyperlipidemia was lower in the EAS ancestry group with a higher risk of hypogonadism (0.98 [0.97-0.98], $p < 10^{-4}$) while risk of hyperlipidemia was higher in the EUR ancestry group with higher risk of hypogonadism (1.01 [1.01-1.02], $p < 10^{-10}$) (Fig. 5B). This differential risk may reflect baseline ancestry-specific difference in metabolic

conditions; previous studies have noted differences in lipid levels in Asian individuals[47–50] Given the number of liver genes linked with variants, we analyzed associations between GS/GRS and different liver disease categories (chronic liver disease, ascites, necrosis). No significant associations were found after multiple test corrections (Supplementary Fig. S10). As expected, higher total testosterone GS were associated with a significantly lower risk of testicular dysfunction, while higher hypogonadism GRS were associated with a higher risk of

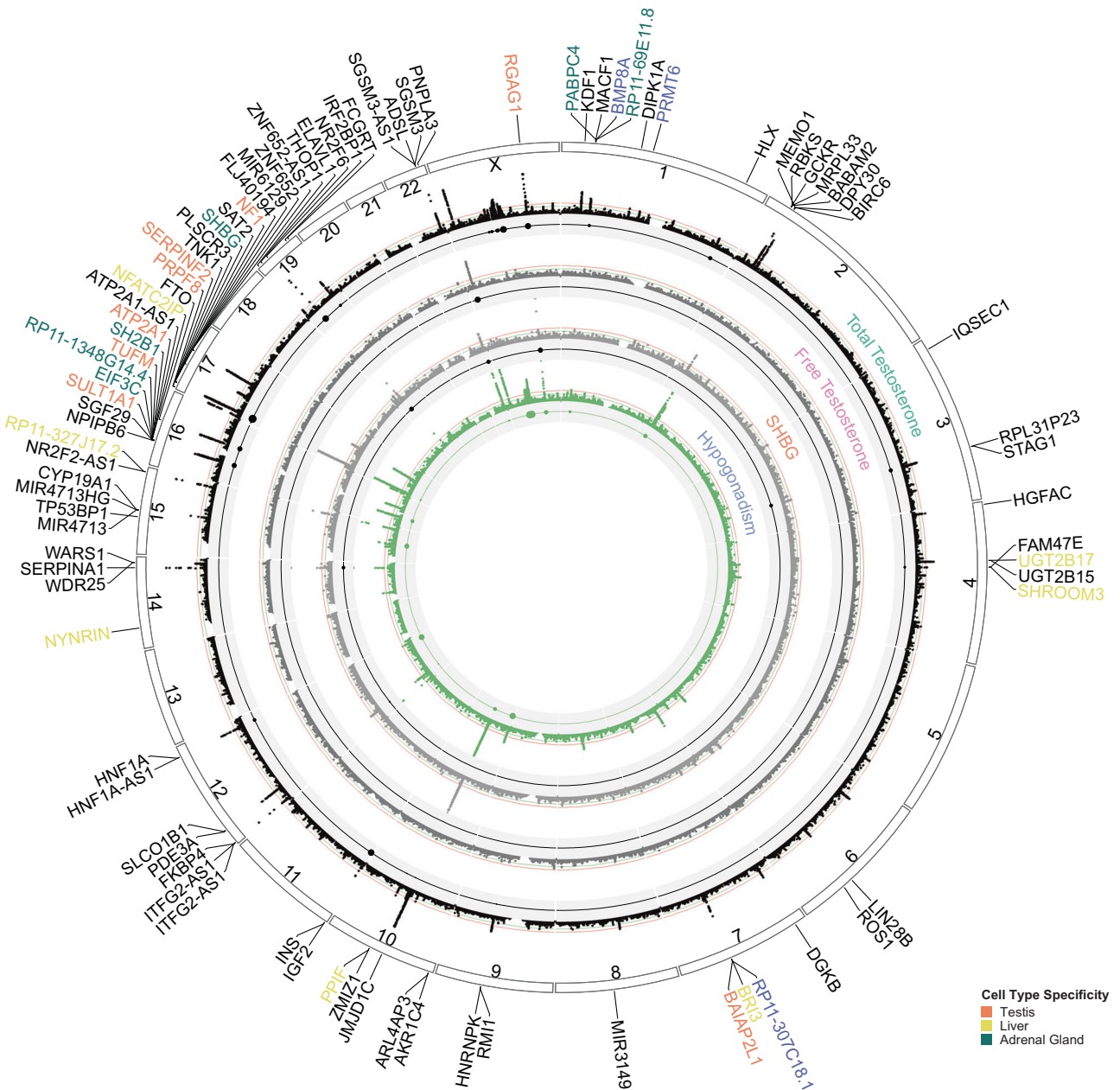

**Fig. 2 | GWAS of Testosterone levels and hypogonadism in MVP men.** Circos plot of METAL *p*-values (-log base 10) of GWAS for total testosterone, free testosterone, SHBG, and hypogonadism in the MVP. The green line indicates a suggestive threshold ($p < 10^{-6}$), and the red line indicates a genome-wide significance threshold ($p < 5 \times 10^{-8}$). The gray band correspond to Cochran's Q statistic to measure heterogeneity. Genes implicated through colocalization analysis (PP H4 > 0.8) are labeled on the outer ring and colored by cell type localization signal: liver (yellow), adrenal gland (green), testes (orange), multiple cell types (purple). A complete list of associations and nearby genes can be found in Supplementary Data 6.

testicular dysfunction in all ancestry groups except the EAS ancestry group (Supplementary Fig. S11A). No significant associations were found between testosterone GS and prostate cancer or dementia (Supplementary Fig. S11B, 11C). These results highlight the role of testosterone and hypogonadism in many diseases.

## Discussion

Utilizing data from the MVP male cohort, we completed a large, multi-ancestral GWAS of total testosterone, free testosterone, SHBG levels, and hypogonadism. We had data for over 10,000 men for each phenotype and identified 157 associations from ancestry-specific GWAS and 188 from MVP meta-analysis. Comparing our findings with the large-scale UK Biobank study by Ruth et al., we found a 64.1% overlap amongst MVP variants associated

with total testosterone, free testosterone, and SHBG ($p < 4.3 \times 10^{-4}$)[26]. After clumping X chromosome variants, the 22 novel variant regions we identified were mostly located on the X chromosome, an important chromosome for androgen receptor regulation. We also identified 9 ancestry-specific signals in men of EUR, AFR, AMR, and EAS ancestry located across many chromosomes (1, 2, 3, 4, 7, 10, 19, X). These results underscore the shared genetic regulation of testosterone and hypogonadism in men, but also unique ancestry-specific associations that may inform future health management and clinical treatment.

Protein-coding variants in several liver genes, such as *PNPLA3, GCKR, SLCO1B1, SERPINA1, HGFAC,* and *UGT2B15,* were significantly associated with total testosterone levels and hypogonadism risk. The *SERPINA1* gene encodes alpha-1 antitrypsin (ATA1), which is a serine

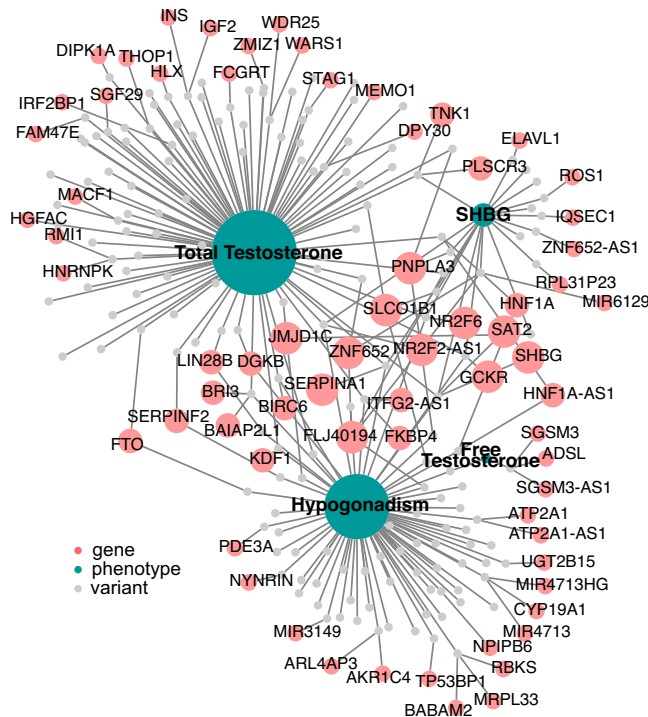

**Fig. 3 | Genes implicated in total testosterone, free testosterone, SHBG and hypogonadism GWAS analysis.** Implication performed using variant effect predictor (VEP). Network plot of genes (coral), variants (gray), and phenotypes (teal). Scatter points are scaled by the number of edges.

**Table 2 | Differences in MVP and UK Biobank populations**

|  | UK Biobank | MVP |
|---|---|---|
| age at enrollment (median [min-max]) | 58 [37–73] | 64 [18–110] |
| % female/male | 54.4/45.6 | 9.2/90.8 |
| % diabetes | 15.1% | 46.6% |
| % hypertensive diseases | 37.0% | 69.5% |
| % lipid disorders | 21.1% | 71.3% |
| % testicular dysfunction | 0.1% | 2.9% |
| % renal failure | 14.1% | 24.5% |
| % liver disease | 5.1% | 19.7% |

protease inhibitor highly expressed in the liver; *NR2F6* encodes a DNA-binding transcription factor that upregulates fatty acid transport and uptake into hepatocytes. *SERPINA1* and *NR2F6* variants have been linked with ALT/AST elevation and MASLD susceptibility[51–55]. *GCKR* and *HGFAC* are also critical for liver glucose utilization and metabolism in the liver[56–58].

Our colocalization analysis further supported a link between liver gene expression and testosterone. *NYNRIN*, a liver enzyme gene, was uniquely identified in colocalization analysis of hypogonadism and gene expression[59]. *UGT2B17*, another hepatic gene, was also identified in the colocalization of SHBG. *UGT2B17* encodes a uridine glucuronosyltransferase, which is involved in testosterone metabolism. In addition, we detected strong colocalization signals for *PRMT6* and *SHBG* in the liver and adrenal gland that were linked to total testosterone and hypogonadism. SHBG is the principal glycoprotein transporting testosterone and synthesized in the liver[10]. SHBG levels have been found to have an inverse relationship with MASLD[58]. PRMT proteins, including PRMT6, have a clinically significant role in hepatic fibrosis and cancer[60]. *PRMT6* encodes arginine methyltransferase that

regulates androgen receptor translocation[61]. Androgen receptors are expressed in both hepatocytes and adrenocortical cells[9]. Our GWAS findings, which demonstrate the overlap between genes linked to testosterone levels and their expression in the liver and adrenal gland, suggest that genetically regulated testosterone may have an important role in normal liver and adrenal function, as well as in hepatic diseases, or that variation in liver and adrenal function can modify testosterone levels and hypogonadism risk.

Using the GS for total testosterone and SHBG and the hypogonadism genetic risk score (GRS) that we developed from our GWAS analyses, we identified associations between testosterone gene variants and the risk of certain diseases, especially metabolic diseases such as diabetes, gout, and hyperlipidemia. Low testosterone levels are associated with increased insulin resistance in men with both type 1 and type 2 diabetes, and insulin sensitivity and testosterone levels are hypothesized to influence one another[62]. The association between testosterone GS and diabetes risk identified in this study is consistent with the findings of Ruth et al., in which a Mendelian randomization analysis suggested that higher testosterone reduced the risk of type 2 diabetes in men[26].

Our study did not find a significant link between higher total testosterone GS and dementia which was unexpected in light of research on the relationship of testosterone and cognitive decline during aging and previous findings that testosterone deficiency may be a risk factor for Alzheimer's disease[21,63–66]. Underlying the memory-enhancing action of testosterone, preclinical research has found that testosterone-activated androgen receptor signaling in the hippocampus can induce synaptic plasticity, promote neuroprotection, and stimulate neurogenesis[67,68]. We also did not find a significant association between prostate cancer and testosterone. Androgen deprivation therapy (ADT) is the mainstay of prostate cancer treatment; however, recent research suggests that there are limits to androgen-related growth[69,70]. Future testosterone GWAS should continue to explore the potential link between testosterone and dementia and prostate cancer, which may involve changes in androgen receptor signaling rather than androgen levels.

A much higher proportion of men with hypogonadism were identified in the MVP (25%) compared to the UK Biobank (0.09%). As a result, we were well-powered to identify significant associations. The high incidence of hypogonadism in MVP may be partially related to the older age of the cohort (64 [18–110] years) compared to the UK Biobank (58 [37–73] years). We found that total testosterone levels decrease with age, with men older than 90 years of age having average total testosterone levels < 300 ng/dl (the threshold for hypogonadism diagnosis). Additionally, it may be related to the higher proportion of chronic disorders that can impact testosterone levels among Veterans, such as diabetes, lipid disorders, and liver disease. Selection bias could also contribute as all participants within the UK Biobank received testosterone testing upon enrollment, while MVP participants only received endocrine testing when ordered within a clinical setting.

The MVP provides a breadth of clinical and genomic data that has allowed us to conduct the most ancestrally diverse genetic analysis of testosterone, SHBG, and hypogonadism to date. However, our study has several limitations. First, as noted above, men participating in the MVP only had testosterone and/or SHBG levels measured when ordered by a physician, likely resulting in a selection bias for individuals already experiencing androgen-related symptoms. We also note limitations with exploring ancestry-specific mechanisms because the available tissue-specific gene expression data is predominantly derived from individuals with EUR ancestry. Thus, our colocalization analyses rely on EUR gene expression data even though gene expression patterns may be influenced by ancestry-specific effects. Furthermore, our GS analysis focuses on phecodes. These clinical outcomes are important for assessing disease risk; however, testosterone can also impact disease severity, treatment response, and disease progression.

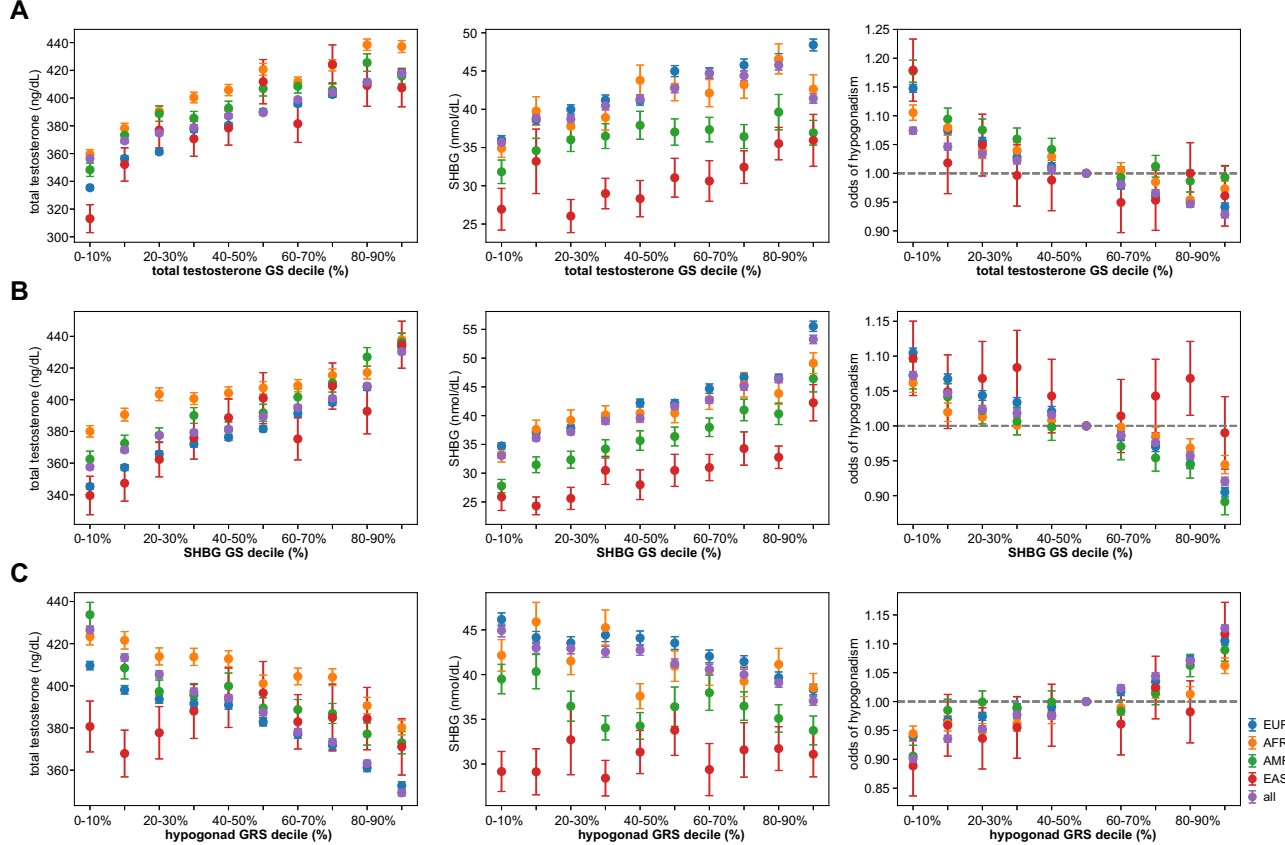

**Fig. 4 | Total testosterone, SHBG GS and hypogonadism GRS construction and internal validation.** Decile plots of total testosterone GS (**A**), SHBG GS (**B**) and hypogonadism GRS (**C**) and corresponding levels of total testosterone (ng/dL), SHBG levels (nmol/dl) and odds of hypogonadism, respectively, all MVP ($n = 658582$) and for EUR ($n = 454996$), AFR ($n = 122068$), AMR ($n = 56513$) and EAS ($n = 10518$) ancestry groups. Data are presented as mean values +/− standard error of the mean (SEM).

Given that over 20% of men over 60 years of age have low testosterone levels[10], this study may help improve clinical care by elucidating the genetic regulation of testosterone and its implication in disease. The Testosterone Trials previously examined the effects of testosterone treatment in older men[71] but were not informed by androgen genetics. Genetic determinants of testosterone levels and cellular actions may provide important insight into testosterone replacement therapy (TRT) action and therapeutic benefit, as well as a greater understanding of androgen effects of testosterone on metabolic, cardiovascular, hepatic, and cognitive function. Moreover, this study deepens our understanding of testosterone genetic regulation and the role of genetically determined testosterone in hypogonadism, diabetes, liver disease, and dementia. Future directions should include the development of improved testosterone-relevant clinical phenotyping from electronic medical records, assessment of interactions between genomic determinants of testosterone and their impact on health and disease, and exploration of the links between testosterone genes and the functions of the liver, cardiovascular and metabolic systems, adrenal cortex, and brain.

## Methods

### Ethical approvals
All MVP participants provided informed consent upon enrollment. This study was approved by the VA Central Institutional Review Board and the Research & Development committees at the Salt Lake City VA and the San Diego VA.

### MVP Genotype quality
All study participants provided blood samples for DNA extraction and genotyping. Blood samples were collected by phlebotomists and banked at the VA Central Biorepository in Boston, MA, where DNA was extracted and shipped to 2 external centers for genotyping. DNA extracted from the buffy coat was genotyped using a custom Affymetrix Axiom Biobank array. The MVP 1.0 genotyping array contains a total of 723,305 variants, enriched for low-frequency variants in African and Hispanic populations and variants associated with diseases common to the VA population[72,73].

### Testosterone and hypogonadism phenotyping
We used the VA Corporate Data Warehouse (CDW) laboratory data to identify Veterans ($N = 702,740$) enrolled in the MVP from fiscal year 2011 to 2018. From the 702,740 Veterans, we selected Veterans who had documentation of at least one hormone laboratory test for testosterone (total testosterone, free testosterone, percent free testosterone, bioavailable testosterone, SHBG, and dihydrotestosterone) ($N = 229,725$). Among these 229,725, only 215,773 were male Veterans with documentation of at least one total testosterone laboratory value. From these 215,773, we excluded those with missing laboratory values ($N = 886$), incorrect units (e.g., mg/dL, ug/dL, %) ($N = 7$), inaccurate logical observation identifiers names and codes (LOINC) ($N = 42$), and a prescription for ADT before the total testosterone laboratory ($N = 4769$) or TRT ($N = 6032$), or had a diagnosis code for testicular cancer ($N = 696$) or orchiectomy ($N = 168$), resulting in a final analytical sample of 203,173 male Veterans (Fig. 1).

We excluded individuals on TRT and ADT to ensure we were assessing endogenous male sex hormone variation. Only morning levels from 7 AM to 12 PM were used. In individuals with multiple readings, the first measurement was used. Abnormally high levels (> 10,000 ng/dL) were removed. We only evaluated men, and patients

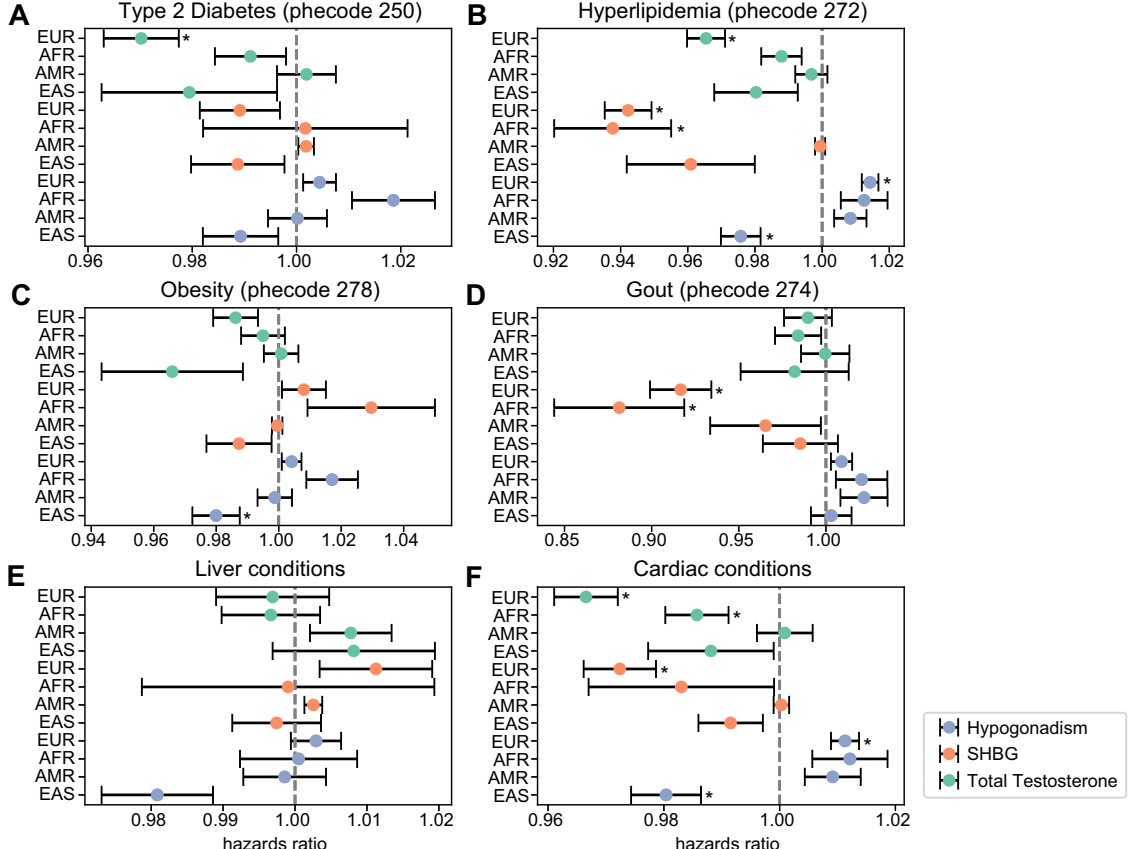

**Fig. 5 | Association of Total Testosterone, SHBG GS and Hypogonadism GRS with Disease Risk in MVP.** CoxPH Hazards ratio of association of total testosterone GS, SHBG GS and hypogonadism GRS with diabetes (**A**), hyperlipidemia (**B**), obesity (**C**), gout (**D**), liver disorders (**E**) and cardiac disorders (**F**) in EUR (total testosterone $n = 292983$, SHBG $n = 373815$, hypogonadism $n = 313728$), AFR (total testosterone $n = 69927$, SHBG $n = 94358$, hypogonadism $n = 75851$), AMR (total testosterone $n = 34304$, SHBG $n = 45567$, hypogonadism $n = 37089$) and EAS (total testosterone $n = 6985$, SHBG $n = 8319$, hypogonadism $n = 7405$) ancestries in the MVP, excluding individuals from discovery analyses. Data are presented as mean values +/− SEM. Asterisk (*) indicates associations significant after Benjamini-Hochberg multiple test correction. Full association statistics are given in Supplementary Data 8.

with sex chromosome aneuploidy (XXY, XYY) were excluded from the analysis.

Individuals with either (1) a documented ICD hypogonadism code or (2) at least 2 low total testosterone values (< 300 ng/dL) were identified as hypogonadal. We selected individuals with no low testosterone readings, TRT, or ADT as matched controls. More detailed information on cohort selection is depicted in Supplementary Fig. S1.

### LDSC
Population-specific LD scores in EUR, AFR, AMR, and EAS ancestry groups were calculated for HapMap[74] populations using intersecting variants from GWAS summary statistics and variant list (w_hm3.snplist, 1,217,312 variants). These scores were then used to calculate heritability estimates for total testosterone, free testosterone, SHBG, and hypogonadism for each ancestry group using LDSC v1.0.1 and default parameters.

### GWAS
Total testosterone, free testosterone, and SHBG levels (ng/dL) were inverse rank normalized. PLINK[75] glm method was applied to conduct association analyses with testosterone levels by each ancestry group (EUR, AFR, AMR, EAS). Ancestry groups were determined by Wendt et al., which combined both a harmonized ancestry and race/ethnicity approach along with random-forest clustering using the 1000 Genomes Project and Human Genome Diversity Project[40]. Variants of minor allele frequency (MAF) less than 0.1% were excluded, and the first ten principal components of ancestry and age were used as

covariates in the analysis. For hypogonadism analysis, the age of recorded hypogonadism diagnosis code or first age of low total testosterone levels (< 300 ng/dL) was used rather than the age of lab test[76]. In individuals with an ICD diagnosis code for hypogonadism and multiple low testosterone readings, the earliest age of hypogonadism identification was used. Significant and independent associations were identified with the linkage- and distance-based clumping method in PLINK using the GWAS significance threshold ($p < 5 \times 10^{-8}$). Variants within 500 kb and with $R^2 > 0.00001$ were pruned away. A strict $R^2$ threshold was used to prepare data for downstream Mendelian randomization results and to ensure independent loci. Alleles with significantly different (population-specific) effect sizes were identified via non-overlapping 90% confidence intervals between the sentinel ancestry and all other ancestries.

For EUR GWAS validation, we used UK Biobank summary statistics for European men (GCST90012103, GCST90012109, and GCST90012113) that were retrieved from Ruth et al.[26] (https://www.ebi.ac.uk/gwas/publications/32042192#study_panel). For significant MVP variants not found in summary statistics, the closest variant (in base pairs) was used as a proxy. Proxy variants used for UK Biobank validation are provided in Supplementary Data 1 and 4. Novel variants were identified as those that met the GWAS significance threshold ($p < 5 \times 0{-8}$) in MVP but not in the UK Biobank.

For AFR GWAS validation, total testosterone, free testosterone, and SHBG values were extracted and inverse-rank normalized for self-identifying African individuals from the Multi-Ethnic Study of Atherosclerosis (MESA). Individuals with KING relatedness > 0.177 were

removed. Also, in cases of family genotypes, parents were excluded from analyses. The PLINK glm method was used to conduct validation analysis of testosterone loci in 1257 men. Age and the first ten principal components of ancestry were used as covariates. We excluded significant MVP total testosterone (1), free testosterone (1), SHBG (1), and hypogonadism (2) variants for the AFR ancestry group, which were not found in summary statistics were excluded from validation analysis.

## METAL

Meta-analysis of MVP EUR, AFR, AMR, and EAS GWAS with total testosterone levels, free testosterone levels, SHBG levels, and hypogonadism risk was conducted with METAL using standard error-based weights. Significant independent loci were identified through the linkage- and distance-based clumping method in PLINK and using GWAS significance threshold ($p < 5 \times 10^{-8}$). Variants within 500 kb and with $R^2 > 0.00001$ were pruned away. Cochran's heterogeneity analysis was conducted to identify variants with high variability between ancestry groups. Values are given in Supplementary Data 4 and plotted in Fig. 2.

## SuSiE Fine-mapping analysis

SuSiE requires a reference LD matrix and GWAS summary statistics. For each significant association, genotypes of all variants within 1 Mb were extracted from each MVP population, thinning individual count to 1000. Individual count had to be thinned to accommodate memory and computing parameters; moreover, LD may be accurately approximated using a subset of individuals from the GWAS cohort. Pairwise linkage disequilibrium was calculated for all variants within loci. For each phenotype (total testosterone, free testosterone, SHBG, hypogonadism), we calculated z-scores by dividing each effect size (beta) value by its standard error. Fine-mapping was run with SuSiE (susie.abf) using $L = 10$, which assumes a maximum of 10 causal variants. Matched reference populations were used for each analysis (e.g., EUR reference population was used for EUR ancestry associations). We focused on variants with a posterior inclusion probability of at least 10%.

## Gene mapping/colocalization analysis

**To map variants to genes we used two approaches.**

1. Variant effect predictor (VEP) was run on all variants. Gene symbol output based on variant location is included in Supplementary Information. The majority of variants without annotation were intergenic or within regulatory regions.
2. Colocalization analysis using coloc R package[43] (version v5.1.0) was used to determine the probability of shared causal variants between testosterone phenotypes and gene expression. We identified variants which were GTEx eQTLs in the European ancestry group (accession date: 10/02/2019) to assemble a gene set of interest. We then obtained full GTEx *cis*-eQTL association results for androgen-regulated tissues and/or sites (liver, testes, adrenal gland). The coloc.abf function was used to conduct analysis between GTEx *cis*-eQTL results and testosterone phenotype association analyses from the MVP EUR ancestry group. Genes with PP.H4 (posterior probability of one common causal variant between studies) of 0.8 between *cis*-eQTL and testosterone phenotype results were reported.

## Disease association analysis

We evaluated independent lead SNPs identified in the GWAS of free testosterone, hypogonadism, SHBG, and total testosterone for association to other diseases (i.e., PheWAS) in the un-analyzed portion of male MVP subjects ($n = 545,764$ for free testosterone, $n = 448,328$ for hypogonadism, $n = 575,922$ for SHBG, and $n = 456,066$ for total testosterone). PheWAS was conducted on 1875 phecode categories[77] with the PheWAS R package[78] (R version 4.0.3) by ancestry group using logistic regression. We restricted the analysis to the male sex and used age and the first ten genotyping principal components as covariates.

To confirm if variants from testosterone analyses shared causal variants with the disease, we extracted variants within a 100 Kb range and MAF of at least 0.1% for the variant of interest. We ran coloc[43] R package (version v3.2.1) on variant *p*-values from total testosterone levels, free testosterone levels, SHBG levels, or hypogonadism GWAS analyses and disease PheWAS. Variant MAF from PheWAS analysis was used in coloc analysis.

## Mendelian randomization

TwoSampleMR (version 0.6.6)[45,46] was run on significant associations for total testosterone, SHBG, and hypogonadism variants from testosterone meta-analysis and published GWAS studies for obesity, hyperlipidemia, gout, type 2 diabetes, and MASLD. Published GWAS details are given in Supplementary Data 7. Variants from MVP meta-analyses were harmonized with published GWAS information. No further clumping was performed as we used strict linkage disequilibrium thresholds to identify significant GWAS variants.

## GS and GRS Construction and disease association

We constructed GS/GRSs using only genome-wide significant meta-analysis variants ($p < 5 \times 10^{-8}$) for total testosterone, SHBG, and hypogonadism. Variants were extracted from imputed genotype BGEN files. Alleles were oriented to increasing testosterone or SHBG levels or increasing the risk of hypogonadism. Genotype dosages were weighted by effect size estimates for each respective ancestry group and summed. Validation of GS/GRS was conducted within discovery groups for each analysis.

Cox proportional hazards analyses were conducted using R survival package (***) in EUR, AFR, AMR, and EAS MVP groups with phecodes used in PheWAS analyses (see Disease Association Analysis) for event and age of diagnosis for time to event. Individuals included in discovery analyses, those with chromosomal abnormalities (XXY, XXX, XYY), and those on ADT or TRT were removed from analyses. Analyses were controlled for age of enrollment and the top 3 genotyping principal components as covariates.

## Reporting summary

Further information on research design is available in the Nature Portfolio Reporting Summary linked to this article.

## Data availability

The full summary-level association data generated in this study are deposited in the GWAS Catalog database https://www.ebi.ac.uk/gwas/ under accession numbers GCST90503313, GCST90503314, GCST90503315, GCST90503316, GCST90503317, GCST90503318, GCST90503319, GCST90503320, GCST90503321, GCST90503322, GCST90503323, GCST90503324, GCST90503325, GCST90503326, GCST90503327, GCST90503328. The source data generated in this study are provided in the Supplementary Datasets provided in this paper.

## Code availability

Code to reproduce manuscript figures is available at: https://github.com/meghatron21/mvp-testosterone-gwas.

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

## Acknowledgements

This research used data from the Million Veteran Program, Office of Research and Development, Veterans Health Administration. This research was supported by the Million Veteran Program MVP022 award CX001727 (PI: Richard L. Hauger MD). This work was supported using resources and facilities of the Department of Veterans Affairs (VA) Informatics and Computing Infrastructure (VINCI), including data analytics conducted by its Precision Medicine research team, which is funded under the research priority to Put VA Data to Work for Veterans (VA ORD 24-D4V-02). This publication does not represent the views of the Department of Veterans Affairs or the United States Government. Dr. Hauger was additionally funded by the VISN-22 VA Center of Excellence for Stress and Mental Health (CESAMH) and National Institute of Aging RO1 grants AG050595 (The VETSA Longitudinal Twin Study of Cognition and Aging VETSA 4) and AG05064 (Effects of Androgen Deprivation Therapy on Preclinical Symptoms of Alzheimer's Disease). Dr. Panizzon was also funded by NIA R01 grants AG050595 and AG05064. Dr. Seibert reports honoraria from Varian Medical Systems and WebMD; he has an equity interest in CorTechs Labs, Inc. and serves on its Scientific Advisory Board; he receives research funding from GE Healthcare through the University of California San Diego. Dr. Lynch, Dr. Teerlink, Dr. Lee, Dr. Chang, and Ms. Pridgen report grants from Alnylam Pharmaceuticals Inc, Astellas Pharm Inc, AstraZeneca Pharmaceuticals LP, Biodesix, Boehringer Ingelheim International GmbH, Celgene Corporation, Eli Lilly and Company, Genentech Inc, Gilead Sciences Inc, GlaxoSmithKline PLC, Innocrin Pharmaceuticals Inc, IQVIA Inc, Janssen Pharmaceuticals Inc, Kantar Health, MDxHealth, Merck & Co. Inc, Myriad Genetic Laboratories Inc, Novartis International AG, Parexel International Corporation through the University of Utah or Western Institute for Veteran Research outside the submitted work. For LDhub analysis, we gratefully

acknowledge all the studies and databases that made GWAS summary data available: ADIPOGen (Adiponectin genetics consortium), C4D (Coronary Artery Disease Genetics Consortium), CARDIoGRAM (Coronary ARtery DIsease Genome wide Replication and Meta-analysis), CKDGen (Chronic Kidney Disease Genetics consortium), dbGAP (database of Genotypes and Phenotypes), DIAGRAM (DIAbetes Genetics Replication And Meta-analysis), ENIGMA (Enhancing Neuro Imaging Genetics through Meta Analysis), EAGLE (EArly Genetics & Lifecourse Epidemiology Eczema Consortium, excluding 23andMe), EGG (Early Growth Genetics Consortium), GABRIEL (A Multidisciplinary Study to Identify the Genetic and Environmental Causes of Asthma in the European Community), GCAN (Genetic Consortium for Anorexia Nervosa), GEFOS (GEnetic Factors for OSteoporosis Consortium), GIANT (Genetic Investigation of ANthropometric Traits), GIS (Genetics of Iron Status consortium), GLGC (Global Lipids Genetics Consortium), GPC (Genetics of Personality Consortium), GUGC (Global Urate and Gout consortium), HaemGen (haemotological and platelet traits genetics consortium), HRgene (Heart Rate consortium), IIBDGC (International Inflammatory Bowel Disease Genetics Consortium), ILCCO (International Lung Cancer Consortium), IMSGC (International Multiple Sclerosis Genetic Consortium), MAGIC (Meta-Analyses of Glucose and Insulin-related traits Consortium), MESA (Multi-Ethnic Study of Atherosclerosis), PGC (Psychiatric Genomics Consortium), Project MinE consortium, ReproGen (Reproductive Genetics Consortium), SSGAC (Social Science Genetics Association Consortium) and TAG (Tobacco and Genetics Consortium), TRICL (Transdisciplinary Research in Cancer of the Lung consortium), UK Biobank. We gratefully acknowledge the contributions of Alkes Price (the systemic lupus erythematosus GWAS and primary biliary cirrhosis GWAS) and Johannes Kettunen (lipids metabolites GWAS).

## Author contributions

Me.S.P., R.L.H., and Ma.S.P. conceived the work and designed the experiments; M.P., N.N.C., R.D., K.M.L., T.A-F., F.Y.A., and J.A.L. assisted in clinical information processing; Me.S.P. performed GWAS and Lab-WAS analysis; C.C.T performed PheWAS analyses; Me.S.P. wrote the paper with assistance from R.L.H, G.K.J., T.M.S., B.S.R., J.A.L., H.K.C., M.P., K.M.P., C.C.T., and T.A.; G.K.J., T.M.S., B.S.R., J.A.L., H.K.C., Ma.S.P., T.A., and R.L.H. advised on genetic analyses and functional/clinical relevance.

## Competing interests

The authors declare no competing interests.
