## [Transparent Peer Review file · Nature Communications]

Discovery of Novel Ancestry Specific Genes for Androgens and Hypogonadism in Million Veteran Program Men

Corresponding Author: Professor Richard Hauger

Version 0:

Reviewer comments:

Reviewer #1

(Remarks to the Author)

This is a well described GWAS of a set of biomarkers important in many conditions - testosterone levels and related measures such as SHBG. I enjoyed reading the paper - and apologies for being late. Importantly it builds on work limited to people of European ancestry by using a study, the Million Vets Population, that is highly diverse. My main comment is specificity of the "ancestry specific" signals - it was not totally clear to me that these might be completely ancestry specific and I think some more diagrams such as those comparing betas could be useful. Otherwise I have mainly minor comments for the authors to consider:

1. The authors point to many signals that are ancestry specific. They defined these as those signals that are GWAS significant after conditioning on the nearest trans-ancestry signal within 500kb, but what if there was more than one statistically independent signal at the locus? Is there evidence that the effect sizes are truly heterogeneous between ancestry groups for robust signals? plots that allow the reader to compare betas would be useful here.
 2. On initial reading 35% of men having hypogonadism sounds incredibly high. Reading further down the paragraph it sounds like this is consistent with the age profile of the cohort, but later on (lines 131 – 140) the comparison to UK Biobank is striking. At baseline the eldest man in UK biobank would have been 70, but are the differences primarily age or due to the enrichment of measures in MVP in men needing testing? but it might be worth elaborating a little on this point?
 3. Line 129 SuSIE causality analysis – worth expanding a little where these 145 lead variants and what does high probability mean in this context? is this a credible set type analysis? Likewise on lines 166?
 4. Lines 144-148 and comparison to SF3. I don't see that there are lots of variants with opposite effects in EAS compared to ancestry groups. Are there some by chance due to the smaller sample size and power in EAS? this is important – a real signal that goes in opposite directions at GWAS significance is highly unusual.
 5. Fig. S5 – consistency in the MESA study – this figure needs 95% CIs added to give a better perspective on the consistency.
 6. Line 179. The co-localisation of an eQTL in an androgen specific tissue and a GWAS signal for one of the hormone levels is certainly promising but doesn't necessarily mean the e-gene is the right one – perhaps clarify the wording here a little when talking about the "36 genes"
 7. Lines 190-196. The PheWAS analysis really needs to be combined with colocalisation – when a testosterone signal is associated with a disease condition, is it the same signal for the disease? there is new software from Chris Wallace's group to perform this in R.
 8. Quoting odds ratios of "(OR=1.00002) seems odd. I guess because this is per allele of the PRS. I would suggest use a std. deviation of the PRS. But see next point
 9. Is the PRS a full genome wide PRS or one that just involves the GWAS significant signals? If the latter then better to term a "genetic risk score", and spell out that it only consists of top signals.
 10. It would help the reader if the discussion could place the findings in context of the previous GWASs of these traits. Several of the genes and many of the loci highlighted were previously identified in the UK Biobank analysis, e.g. SERPINA, and many others.
 11. In the methods what does "we first mapped genes via VEP" mean? what if the variant was not annotated with a gene name? what if two signals were close by but annotated as different genes? I would suggest this analysis would be more robust if performed by chromosome and position rather than annotation.
- thanks Tim Frayling

Reviewer #2

(Remarks to the Author)

This paper describes a GWAS of total testosterone (n=139,226 men), SHBG (49,709), and free testosterone (19,635) in the Million Veterans Programme (MVP) including both transancestry and ancestry specific analyses in European, African, Admixed American, and East Asian ancestry groups. They find 164 significant trans-ancestry associations, of which 9 were novel, and (a further) 102 ancestry-specific associations (unclear how many of these are novel). Using routinely generated (clinically indicated) measurements, 35% had a low total testosterone levels and ~25% met criteria for hypogonadism.

Major comments:

1) There are many instances of vague statements which require more detail:

Page 5: "Of the significant associations in the MVP, 54 (86%), 13 (100%), and 1 (100%) variants are significant in UK Biobank ($p < .05/164$) for total testosterone, SHBG, and free testosterone, respectively." Add the denominators and explain why the numbers seem so much lower than the total numbers of signals found in MVP.

Discussion Page 8 "Several trans-ancestry variants were validated using the large-scale UK Biobank study by Ruth et al". Add here how many (and %) overall are validated.

Page 5 "Effects of Genetic Variants Influencing Male Sex Hormones are Consistent across Ancestry Groups". As well as citing Fig S3, the main text should describe the findings quantitatively. How many (N, %) were concordant / discordant?

Page 6 "beta values of associations are consistent with MVP associations (Figure S5)" as well as citing Fig S5 the main text should describe the findings quantitatively.

2) Validation of MVP signals: is done only for the transancestry signals in UK Biobank and the African American men in MESA.

The 65 EUR-specific signals should be validated in UK Biobank by Ruth et al.

Page 8 - "Nine total testosterone variants were uniquely implicated by our MVP multiancestry study." Do you mean 9 signals from the transancestry analysis. I didn't see comparison with UK Biobank for the EUR-specific variants.

3) Links to other diseases seem to be oddly selected for testing.

Page 8: PGS analysis: "These results highlight the multitude of roles for testosterone in human disease" - this statement is supported by sparse and inconsistent results. Phewas identified joint associations between male hormone levels with gout, lipids, diabetes (presumably Type 2 ?), CVD, hypertension, IHD, PSA and prostate cancer. To justify the above claim, PGS analysis should be performed not only for diabetes, and why also choose dementia? Also add PGS analysis using SHBG.

Page 8 - "These variants also had implications for ... skin cancer". This must be wrongly included here. Skin cancer is not mentioned elsewhere.

Page 9: Genes containing protein coding signals are discussed at some length. How many of the listed associated missense variants are novel? This section largely discusses overlap with liver disease at SERPINA1, GCKR and PNPLA3. Why is liver disease/ALT not highlighted in the Phewas analysis? Can they deduce whether higher testosterone is genetically associated with higher or lower risk of ALT elevation / NAFLD?

4) Page 10: Testosterone and dementia. This section doesn't mention the opposite direction association in Admixed Americans (higher testosterone and increased odds of dementia). Is there any evidence for ancestry differences in causes of dementia, e.g. Alzheimers, hypertension / stroke related?

5) Page 8: "These (novel) variants were primarily located on the X-Chr". Is there any reason for this? Could this indicate incomplete conditional analysis? It would be helpful to list these 9 specifically in a table indicating Chr positions etc.

6) Figure 5B - there is something drastically wrong here because the estimate for 'all' is much lower than in each of the ancestry groups separately.

Other comments:

Page 5 "Effects of Genetic Variants Influencing Male Sex Hormones are Consistent across Ancestry Groups" - this is an odd title considering the text highlights that "Variant effects are consistently different in the East Asian group" - and is inconsistent with the next title on Page 6. But is this because of the very small sample size of East Asians? Is there actual statistical heterogeneity?

Page 8 - add main text citation for Fig S7

Figure 2: legend: clarify that gene implication was performed using GTEx cis-eQTL gene expression information

Figure 4: title "Polygenic Score Construction and Validation for..." - clarify that this is simply "internal validation" as this figure does not include additional non-discovery data.

Version 1:

Reviewer comments:

Reviewer #1

(Remarks to the Author)

The authors have responded to comments but I still feel the manuscript needs a lot of tightening up. It is worth asking someone with extensive GWAS experience to have a good read and check. I sense it was written by experts in the phenotypes rather than the genotypes. Here are some things that may help:

1. In the abstract its important to reword to make clear you've not found genes but variants . This is also important in the main text - rather than refer to associations with genes (intro for example) I suggest « associations of variants in or near genes »
2. The wording of the Susie analysis page 4 and 5 is a little odd. What do the authors mean when they claim an association to be causal ? Do they mean that the PIP evidence is string that the lead variant is the causal variant ? If so I suggest it is better to talk about the « causal variant »
3. Likewise the wording is confusing in other places for example « Of the significant associations in the MVP, 63 total testosterone, 13 SHBG and 1 free testosterone associations were found in UK Biobank summary statistics, respectively » I think the authors mean the variant was available for look up not that the association was found (the « association was found » implies that the association between the variant and testosterone levels was replicated .
4. Of the 164 variants found to be associated with sex hormone levels in MVP, less than half were available in the UK Biobank data which seems strange - the full GWAS stats from Ruth et al. Are available on line at the EBI GWAS catalogue and close LD proxies could be used if specific rsIDs are not available .
5. Again on lines 169-170 the wording is confusing - « associations were found » implies replication not just that the variant was present
6. Lines 237 onwards - if the dementia result did not reach bonferroni significance I suggest do not mention it could raise expectations .
7. Lines 286 in the discussion . Is it possible the « prevalence » of hypogonadism in MVP is not a true reflection of the population prevalence but a feature of clinical testing of men suspected of having low levels ? I realise you talk about this but in places you suggest the differences with the uk Biobank could be due mainly to age .

Reviewer #2

(Remarks to the Author)

The paper is overall much clearer and specifically the authors have significantly improved the level of detail provided in the text to support their stated results. I have 2 further comments.

1. The more extensive PGS-to-disease analysis now provides much greater substantiation for the claimed "multitude of roles for testosterone in human disease" as the PGS provides much stronger evidence for a causal effect of testosterone than the PheWAS which tests individual SNPs. While they took a logical approach to choose the disease outcomes for PGS testing ("We focused on top 50 significant associations from PheWAS analysis which included diabetes, hyperlipidemia, gout, liver disease, and obesity"), unfortunately this excludes some outcomes of high clinical interest and which were highlighted by missense variant associations. The most important of these is prostate cancer (or PSA as a proxy) which should be added to the PGS analysis and also possibly osteoporosis. While the paper finds overall possible benefits for testosterone on metabolic health outcomes, prostate cancer is the main potential adverse effect.
2. Testosterone and dementia. In response to my previous comment, the authors have responded that these associations were not significant after multiple testing (Lines 238-240). The relevant discussion should therefore be appropriately guarded (279-285) as it currently implies that a clear result was found.

Version 2:

Reviewer comments:

Reviewer #1

(Remarks to the Author)

The authors have tightened up the manuscript. I have some final very small comments:

1. Table S3 is in an incomprehensible pdf format. I strongly suggest to reformat all tables into spreadsheets so that readers can easily scan all the information, if this is not a journal requirement I suggest this will help readability and citability !
2. almost all of the associations identified were found in the UKBiobank, and the authors claim "and four total testosterone and two free testosterone testosterone 139 associations were novel" but I cannot see any information about these 6 variants in the tables. These novel findings should be highlighted in a table, but also the UKB data for these 6 needs to be shown, could they be false positives ?
4. line 258 discussion . The authors claim to have "discovered" associations between testosterone gene variants and diseases but some of these were described in Ruth et al. ?

Reviewer #2

(Remarks to the Author)

Lines 194-197 & Figure 4 - specify 'decile' rather than 'quantile' which too vague. Also, the text wrongly describes the 'bottom quantile' as the comparator - in Figure 4 it appears to be the 6th decile.

Lines 203-208 & Figure S8 - for Mendelian randomisation associations, clarify which exposure set was used, i.e. which ancestry or was it from the multi-ancestry meta-analysis? (in the preceding para ancestry specific GRS are used).

Cox proportional hazards analyses - clearly describe what is the multiple test significance threshold used here? The authors previously stated that associations with dementia did not reach this threshold. But now they describe an association with chronic liver disease at " $p < .04$ ".

The lack of association with prostate cancer is surprising. Are numbers of cases of disease in the analysed MVP sample shown (I could not find it)? Might these be insufficient?

REVIEWER COMMENTS

Reviewer #1 (Remarks to the Author): Tim Frayling

This is a well described GWAS of a set of biomarkers important in many conditions - testosterone levels and related measures such as SHBG. I enjoyed reading the paper - and apologies for being late. Importantly it builds on work limited to people of European ancestry by using a study, the Million Vets Population, that is highly diverse. My main comment is specificity of the "ancestry specific" signals - it was not totally clear to me that these might be completely ancestry specific and i think some more diagrams such as those comparing betas could be useful. Otherwise I have mainly minor comments for the authors to consider:

1. The authors point to many signals that are ancestry specific. They defined these as those signals that are GWAS significant after conditioning on the nearest trans-ancestry signal within 500kb, but what if there was more than one statistically independent signal at the locus? Is there evidence that the effect sizes are truly heterogeneous between ancestry groups for robust signals? plots that allow the reader to compare betas would be useful here.

We appreciate the reviewer's positive feedback. To address the reviewer's comment on the ancestry-specific signals, we first determined whether the ancestry-specific signals were significant in trans-ancestry analysis. This removes any variability in results from conditioning on a nearby trans-ancestry signal if there are multiple independent signals in the locus. Of the 157 variants identified through ancestry-specific GWAS of total testosterone, free testosterone, SHBG levels and hypogonadism, 100 variants were significant in trans-ancestry analyses ($p < 5 \times 10^{-8}$).

We next determine if effect sizes were truly heterogenous between ancestry groups. Of the 57 variants not significant in trans-ancestry analysis, we identified 13 ancestry-specific signals with different effect sizes compared to other ancestry group beta values (beta +/- SE). As suggested by the reviewer, we have included figures for ancestry specific signals to allow reader to compare betas. We included examples from each category (ancestry-specific signal significant in trans-ancestry GWAS, ancestry-specific signal with similar beta values across groups, ancestry-specific signal with significantly different beta values) as **Figure S5**.

Figure S5: Variant Effect Sizes from MVP Ancestry Group GWAS. Plots of ancestry-specific variant effect sizes which are significant in trans-ancestry analysis (**A**), which are not significant in trans-ancestry analysis yet with similar beta values with other ancestry groups (**B**), and which are not significant in trans-ancestry analysis and with significantly different beta values with other ancestry groups (**C**). Variants in (**C**) are ancestry-specific signals.

2. On initial reading 35% of men having hypogonadism sounds incredibly high. Reading further down the paragraph it sounds like this is consistent with the age profile of the cohort, but later on (lines 131 – 140) the comparison to UK Biobank is striking. At baseline the eldest man in UK biobank would have been 70, but are the differences primarily age or due to the enrichment of measures in MVP in men needing testing? but it might be worth elaborating a little on this point?

We have included in Table 2, a side-by-side comparison of age and incidence of chronic diseases in the MVP compared to UK Biobank. The increased incidence of hypogonadism in the MVP may be due to an overall older population (64 [18-110]) compared to the UK Biobank (58 [37-73]). In Figure S2D-F, we show decreased total

testosterone levels with age in the MVP, with nearly all men age > 90 with total testosterone levels < 300 ng/dL (hypogonadism diagnosis total testosterone threshold). These findings indicate an older study population would thus have increased diagnosis of hypogonadism. Additionally, there is increased incidence of several chronic diseases, such as diabetes, lipid disorders, and liver disease, which may impact testosterone levels and hypogonadism diagnosis. Since total testosterone, SHBG, and free testosterone levels are lab tests which had to be ordered by a physician, it is possible that this enrichment of chronic disorders may have prompted more testosterone testing and thus hypogonadism diagnosis.

Table 2: Differences in Million Veteran Program (MVP) and UK Biobank populations.

	UK Biobank	MVP
Age at Enrollment (median [min-max])	58 [37-73]	64 [18-110]
% female/male	54.4/45.6	9.2/90.8
% diabetes	15.1%	46.6%
%hypertensive diseases	37.0%	69.5%
%lipid disorders	21.1%	71.3%
%testicular dysfunction	0.1%	2.9%
%renal failure	14.1%	24.5%
%liver disease	5.1%	19.7%

We have included a paragraph in the discussion to elaborate on this difference in hypogonadism diagnosis.

3. Line 129 SuSiE causality analysis – worth expanding a little where these 145 lead variants and what does high probability mean in this context? is this a credible set type analysis? Likewise on lines 166?

Although we found 164 associations, 14 variants were associated with multiple phenotypes. For example, variant 1:26958422:C:T was significantly associated with both hypogonadism and total testosterone. Thus, we identified 145 unique significant variants.

We have expanded on the SuSiE analysis in the manuscript to include a definition of “high probability.” High posterior inclusion probability (PIP) describes how likely associations are to be causal. Per previous research, variants with PIP > 0.10 tend to be causal. However, we also included % of associations to be casual using more relaxed threshold of PIP > 0.05.

“We performed SuSiE causality analysis of 164 associations. An important advantage of SuSiE is that it allows for multiple causal variants in a region, in contrast to other causality inference methods. Using a posterior inclusion probability (PIP) of 0.10, 47 (28.7%) associations were causal. Using a less stringent PIP of 0.05, 52 (31.7%) associations were likely causal. A higher fraction of causal associations versus non-causal associations were missense variants (20.8% vs 4.5%).”

4. Lines 144-148 and comparison to SF3. I don't see that there are lots of variants with opposite effects in EAS compared to ancestry groups. Are there some by chance due to the smaller sample size and power in EAS? this is important – a real signal that goes in opposite directions at GWAS significance is highly unusual.

We have revised **Figure S3** to better visualize effect sizes and significance of these effects in different ancestry groups. Indeed, effect sizes that are in the opposite direction in EAS, such as chrX:62521984:T:C for total testosterone, are not significant and thus unlikely to be real signals.

In the main text, we have revised the response to clarify that most effects are similar across ancestry groups. Only 4 associations (2.4%) have discordant beta values where associations with opposite effects are significant.

Figure S3: Effects of Total Testosterone, SHBG, and Hypogonadism Risk Variants Across MVP Ancestry Groups. Clustermap of total testosterone (A), hypogonadism (B) and SHBG (C) variant effect sizes (beta) for EUR, AFR, AMR, and EAS ancestry groups. Free testosterone not shown as only 1 trans-ancestry variant was identified.

5. Fig. S5 – consistency in the MESA study – this figure needs 95% CIs added to give a better perspective on the consistency.

We have updates Figure S5 (now Figure S6) to include standard error values from Testosterone GWAS in MESA and MVP. As expected, MESA standard error values are higher as sample size is much smaller compared to MVP.

Figure S6: Validation of MVP AFR GWAS variants in MESA cohort. Scatterplot of MVP and MESA Testosterone GWAS beta and standard error values in only AFR men. No points pass significance threshold (5×10^{-8}).

6. Line 179. The co-localisation of an eQTL in an androgen specific tissue and a GWAS signal for one of the hormone levels is certainly promising but doesn't not necessarily mean the e-gene is the right one – perhaps clarify the wording here a little when talking about the “36 genes”

We have removed the term e-gene and instead clarified interpretation of genes with high colocalization signal:

“In total, we discovered 28 genes with high probability of shared causal variants between their expression and total testosterone levels (25), hypogonadism (16) and SHBG levels (11).”

*the 36 genes accounts for both trans-ancestry and ancestry-specific gene associations. However, we included only trans-ancestry analysis results in main text to be more clear.

7. Lines 190-196. The PheWAS analysis really needs to be combined with colocalization – when a testosterone signal is associated with a disease condition, is it the same signal for the disease? There is new software from Chris Wallace's group to perform this in R.

Based on reviewer's suggestion, the goal is to determine if causal variants for testosterone phenotypes are shared with causal variants for disease condition. Per recommendation of Chris Wallace, coloc is best suited to determine shared signals. We ran coloc for 1773 significant PheWAS associations. Of these, 1145 (64.6%) had high probability of shared causal variants between disease condition and testosterone phenotype. We have included this detail in the manuscript.

“To determine whether PheWAS associations were due to testosterone changes, we ran colocalization analysis to determine probability of shared causal variants between testosterone and disease. Approximately 64.6% of the associations had a greater than 80% probability of shared casual variants.”

8. Quoting odds ratios of “(OR=1.00002) seems odd. I guess because this is per allele of the PRS. I would suggest use a std. deviation of the PRS. But see next point

We agree with the reviewer here. We had validated both total testosterone and SHBG genetic scores (GSs) in the UK Biobank. However, hypogonadism genetic risk score (GRS) did not seem to stratify risk of hypogonadism in UK Biobank (**Figure S7C**), possibly due to a low incidence of hypogonadism in the UK Biobank (0.09%).

In a logistic regression of hypogonadism ICD10 code and hypogonadism GRS, age at enrollment, and top 3 principal components in the UK Biobank, the association of GRS with hypogonadism diagnosis was significant yet had a tiny effect size (coef= 2.3×10^{-5} , SE= 1.1×10^{-5} , $P < .03$).

Given these reasons, we have removed the logistic regression analysis and instead conclude that hypogonadism GRS does not seem to stratify odds of hypogonadism in UK Biobank.

9. Is the PRS a full genome wide PRS or one that just involves the GWAS significant signals? If the latter then better to term a “genetic risk score”, and spell out that it only consists of top signals.

The PRS only includes significant trans-ancestry variants. To address reviewer’s comment, we have now modified manuscript to use the better term of genetic risk score (GRS) or genetic score (GS). We have also highlighted in the methods that we only used GWAS significant signals.

Results:

“Given the number of disease associations with identified testosterone variants, we constructed GSs/GRSs including significant trans-ancestry associations for each phenotype”

Methods:

“We constructed GS/GRSs using only significant trans-ancestry variants ($p < 5 \times 10^{-8}$) for total testosterone, SHBG and hypogonadism.”

10. It would help the reader if the discussion could place the findings in context of the previous GWASs of these traits. Several of the genes and many of the loci highlighted were previously identified in the UK Biobank analysis, e.g. SERPINA, and many others.

We have added more description of previous GWAS of testosterone in the manuscript. We specifically included description of genes implicated by shared variants:

“*SERPINF2*, *PRPF8*, *PABPC4*, *BRI3*, *BAIAP2L1*, *SHBG*, *PRMT6*, *RGAG1*, *RP11-307C18.1*, *RP11-327J17.2* and *PPIF* have significantly shared colocalization signal with total testosterone, SHBG levels and hypogonadism. In the GTEx cis-eQTL gene expression profile in the liver, *NYNRIN* was uniquely identified by colocalization analysis of hypogonadism but not total testosterone, while *RGAG1* and *UGT2B17* were uniquely identified through colocalization analysis of SHBG but not total testosterone. Results from colocalization analysis were compared with cell-type analysis in UK Biobank by Ruth et al. *UGT2B17* was implicated by liver eQTL analysis of both SHBG and total testosterone levels. *BRI3* and *PRMT6* were liver eQTLs in total testosterone analysis while *PPIF* was a liver eQTL in SHBG analysis. *PABPC4* was a muscle eQTLs in SHBG analysis.”

11. In the methods what does “we first mapped genes via VEP” mean ? what if the variant was not annotated with a gene name ? what if two signals were close by but annotated as different genes ? I would suggest this analysis would be more robust if performed by chromosome and position rather than annotation.

We used two methods to implicate genes.

1. VEP uses variant chromosome and position to determine nearby genes the variant could modify. 81 of the 164 trans-ancestry associations were not annotated with a gene name if they were intergenic variants or located in a regulatory regions that do not map genes. Since this is just an annotation, we wanted to complement it with a more robust colocalization analysis.
2. For the colocalization analysis, we identified any variants which were GTEx eQTLs. We ran coloc R package to determine if there were shared causal variants between GTEx eQTL and testosterone phenotype analyses.

We have included results of both analyses in the manuscript and gave more clarification about the gene mapping in the methods.

Reviewer #2 (Remarks to the Author):

This paper describes a GWAS of total testosterone (n=139,226 men), SHBG (49,709), and free testosterone (19,635) in the Million Veterans Programme (MVP) including both transancestry and ancestry specific analyses in European, African, Admixed American, and East Asian ancestry groups. They find 164 significant trans-ancestry associations, of which 9 were novel, and (a further) 102 ancestry-specific associations (unclear how many of these are novel). Using routinely generated (clinically indicated) measurements, 35% had a low total testosterone levels and ~25% met criteria for hypogonadism.

Major comments:

- 1) There are many instances of vague statements which require more detail:

We thank the reviewer fo highlighting vague statements that we have addressed below. We also have reviewed the manuscript to revise other vague statements. For example, there were cases where we did not specify if we were referring to trans-ancestry or ancestry-specific variants in gene colocalization and PheWAS analysis. We have now described results specific to trans-ancestry analysis and modified the main text:

To identify candidate genes influencing testosterone levels, we performed colocalization analysis using GTEx cis-eQTL gene expression information for androgen-regulated tissues and/or sites and trans-ancestry GWAS results. The colocalization analysis estimates the probability of shared causal signal between cis-eQTL and GWAS analyses of total testosterone, free testosterone, SHBG levels and hypogonadism.⁴² In total, we discovered 28 genes with high probability of shared causal variants between their expression and total testosterone levels (25), hypogonadism (16) and SHBG levels (11)

Twenty-six trans-ancestry variants have significant disease associations. Many variants were associated with testicular hypofunction and dysfunction (n=8), gout (n=11), lipid metabolism (n=9), type 2 diabetes (n=8), and cirrhosis (n=6).

Page 5: "Of the significant associations in the MVP, 54 (86%), 13 (100%), and 1 (100%) variants are significant in UK Biobank ($p < .05/164$) for total testosterone, SHBG, and free testosterone, respectively." Add the denominators and explain why the numbers seem so much lower than the total numbers of signals found in MVP.

To identify significant association overlapping with UK Biobank analyses, we used summary statistics from Ruth et al. Within our MVP analyses, 76, 73, 14 and 1 significant associations were identified for hypogonadism, total testosterone, SHBG and free testosterone, respectively. However, in the Ruth et al. UK Biobank summary statistics, only 63, 13 and 1 significant associations for total testosterone, SHBG, and free testosterone,

respectively, were found. We evaluated shared variants between both MVP and UK Biobank. We have clarified this further in the manuscript.

Of the significant associations in the MVP, 63 total testosterone, 13 SHBG, and 1 free testosterone associations were found in UK Biobank summary statistics, respectively. For the variants found to be overlapping in MVP and UK Biobank, 54 (86%), 13 (100%), and 1 (100%) variants were significant in UK Biobank ($p < .05/164$) for total testosterone, SHBG, and free testosterone, respectively.

Discussion Page 8 "Several trans-ancestry variants were validated using the large-scale UK Biobank study by Ruth et al". Add here how many (and %) overall are validated.

We have updated the discussion to specify how many variants were validated:

Comparing our findings with the large-scale UK Biobank study by Ruth et al., we found 88.3% overlap amongst MVP variants associated with total testosterone, free testosterone and SHBG.

Page 5 "Effects of Genetic Variants Influencing Male Sex Hormones are Consistent across Ancestry Groups". As well as citing Fig S3, the main text should describe the findings quantitatively. How many (N, %) were concordant / discordant?

We have updated the main text to quantify the findings. Of the 164 trans-ancestry associations, only 4 variants (2.4%) have discordant beta values. Three associations were with total testosterone and the other was with hypogonadism. Association with hypogonadism (X:68264549:A:G) was significantly different in EUR and AFR ancestry group compared to trans-ancestry group. Two total testosterone associations (1:120117890:G:C, 19:17236045:G:C) had significantly different effect sizes in EAS ancestry group compared to trans-ancestry group.

Page 6 "beta values of associations are consistent with MVP associations (Figure S5)" as well as citing Fig S5 the main text should describe the findings quantitatively.

We have revised the main text and figure to include pearson correlation of beta values, which show significant positive correlation (pearson $r = 0.64$, $p < .005$).

2) Validation of MVP signals: is done only for the transancestry signals in UK Biobank and the African American men in MESA.

The 65 EUR-specific signals should be validated in UK Biobank by Ruth et al.

We have now included quantification of EUR GWAS signals which were validated by Ruth et al.

Of the 98 associations identified by GWAS in the EUR population, 34, 7, and 1 MVP associations for total testosterone, SHBG, and free testosterone were found in the UK Biobank, respectively. Of these, 32 (94.1%), 7 (100%), and 1 (100%) were validated in the UK Biobank ($p < .05/98$).

Page 8 - "Nine total testosterone variants were uniquely implicated by our MVP multiancestry study." Do you mean 9 signals from the transancestry analysis. I didn't see comparison with UK Biobank for the EUR-specific variants.

We have now clarified that the nine novel total testosterone variants were from trans-ancestry analysis. Based on additional X chromosome analysis suggested by the reviewer, we have 6 novel trans-ancestry variants (3 variants were not significant after conditional X chromosome analysis).

We also now have included validation of EUR variants in UK Biobank (See previous comment)

3) Links to other diseases seem to be oddly selected for testing.

Page 8: PGS analysis: "These results highlight the multitude of roles for testosterone in human disease" - this statement is supported by sparse and inconsistent results. PheWAS identified joint associations between male hormone levels with gout, lipids, diabetes (presumably Type 2 ?), CVD, hypertension, IHD, PSA and prostate cancer. To justify the above claim, PGS analysis should be performed not only for diabetes, and why also choose dementia? Also add PGS analysis using SHBG.

We have now given more description of selection of human diseases. We focused on top 50 significant associations from PheWAS analysis which included diabetes, hyperlipidemia, gout, liver disease, and obesity. We first ran Mendelian Randomization on total testosterone, SHBG and hypogonadism associations and publicly available GWAS of diabetes, hyperlipidemia, gout, liver disease, and obesity (**Figure S8A**). We found increased risk of gout (beta=0.35, p<0.01) and NAFLD (beta=0.18, p<.06) in individuals with higher risk of hypogonadism. We also found higher levels of SHBG were causally associated with lower risk of gout (beta=-1.15, p<.0009) and hyperlipidemia (beta=-0.002, p<.03).

Figure S8: Mendelian Randomization (MR) of Total Testosterone, SHBG and Hypogonadism and Disease Incidence. (A) Plot of MR effect sizes for total testosterone, SHBG, and hypogonadism and gout, type 2 diabetes, obesity, hyperlipidemia, and NAFLD.

Given these findings, we proceeded to evaluate the associations of our constructed and validated genetic (GS) and genetic risk scores (GRS) for total testosterone, SHBG, and hypogonadism with these metabolic and liver conditions (**Figure 5**). Higher total testosterone levels were associated with lower risk of type 2 diabetes and hyperlipidemia in EUR (0.93 [0.92-0.94], p<10⁻⁸) and AFR (0.94 [0.93-0.95], p<10⁻⁵) ancestry groups. Higher SHBG levels were associated with lower risk of gout in EUR (0.92 [0.91-0.93], p<10⁻⁷) and AFR (0.91 [0.89-0.93], p<10⁻⁵) ancestry groups and higher risk of liver disorders, such as hepatomegaly, liver necrosis, and abnormal liver function tests, in EUR (1.05 [1.04-1.07], p<10⁻³) and AMR (1.18 [1.15-1.18], p<10⁻⁵) ancestry groups. Risk of hyperlipidemia was lower in the EAS ancestry group (0.98 [0.97-0.98], p<10⁻⁴) with higher hypogonadism risk scores compared to higher risk of hyperlipidemia in the EUR (1.02 [1.01-1.02], p<10⁻¹⁰) ancestry group with higher hypogonadism risk scores.

Figure 5: Association of Total Testosterone, SHBG GS and Hypogonadism GRS with Disease Risk in MVP. CoxPH Hazards ratio of association of total testosterone GS, SHBG GS and hypogonadism GRS with diabetes (A), hyperlipidemia (B), obesity (C), gout (D), and liver disorders (E) in all, EUR, AFR, AMR and EAS ancestry in the MVP.

Page 8 - "These variants also had implications for ... skin cancer". This must be wrongly included here. Skin cancer is not mentioned elsewhere.

We removed the mention of skin cancer and confirmed significant PheWAS analysis associations.

Page 9: Genes containing protein coding signals are discussed at some length. How many of the listed associated missense variants are novel? This section largely overlaps with liver disease at SERPINA1, GCKR and PNPLA3. Why is liver disease/ALT not highlighted in the PheWAS analysis? Can they deduce whether higher testosterone is genetically associated with higher or lower risk of ALT elevation / NAFLD?

Of the missense variants, only rs701564 (HLA-DQB1 missense variant) is novel when compared to the UK Biobank. Missense variants were implicated in many conditions, such as liver disease, diabetes, hyperlipidemia, ischemic heart disease, gout, osteoporosis, and elevated prostate-specific antigen (PSA).

We have now revised manuscript to highlight several significant associations with liver function (phecode 571, 573). Several variants (6) were associated with cirrhosis. Additionally, of the trans-ancestry variants, 38 (26.2%) and 42 (29.7%) were significantly associated with median ALT and AST levels, respectively.

In order to determine the role of testosterone with NAFLD, we conducted Mendelian Randomization. We found increased risk of NAFLD in individuals with hypogonadism.

Figure S8: MR of Total Testosterone, SHBG and Hypogonadism and Disease Incidence. (A) Plot of MR effect sizes (beta) for total testosterone, SHBG, and hypogonadism and gout, type 2 diabetes, obesity, hyperlipidemia, and NAFLD.

4) Page 10: Testosterone and dementia. This section doesn't mention the opposite direction association in Admixed Americans (higher testosterone and increased odds of dementia). Is there any evidence for ancestry differences in causes of dementia, e.g. Alzheimers, hypertension / stroke related?

The opposite direction could be due to biological differences or due to confounding variables. In order to ensure association was not due to confounding variables, we revised our methods to run Cox proportional hazards. CoxPH accounts for age of diagnosis, thus better suited for analysis of chronic conditions that are dependent of age. We also removed individuals from discovery analyses, any individuals with chromosomal abnormalities, and any individuals on androgen deprivation therapy (ADT) or testosterone replacement therapy (TRT). Results from analysis show different effect sizes in association of total testosterone and dementia in African and Admixed American ancestry group; however, these associations are not significant.

Figure S9: Association of Total Testosterone, SHBG GS and Hypogonadism GRS with Testicular Dysfunction and Dementia Risk. CoxPH Hazards ratio of total testosterone GS, SHBG GS and hypogonadism GRS and testicular dysfunction (A) and dementia (B) in all, EUR, AFR, AMR, EAS ancestry groups in MVP.

5) Page 8: "These (novel) variants were primarily located on the X-Chr". Is there any reason for this? Could this indicate incomplete conditional analysis? It would be helpful to list these 9 specifically in a table indicating Chr positions etc.

variant_id	SNP	P	pheno	A1	BETA	SE	consequence	gene	susie	phenotype	conditional X varia	conditional p
rs4846670	1:220895675:G:T	4.33E-08	total	T	0.0232856	0.00425151	regulatory_region_variant,int	-	0.13125713			
rs701564	6:32662032:C:T	5.18E-09	total	T	-0.0275719	0.00471988	missense_variant,downstream	HLA-DQB1,H	NA			
rs7890685	X:50772588:C:T	2.62E-09	total	C	-0.0324048	0.00544249			0.51275934	Diabetes mellitus,Type 2 diabetes,Obesity,Act	X:64823258:T:C	8.74E-04
rs16991372	X:64823258:T:C	3.00E-12	total	C	-0.0461227	0.0066093			0.42637849	Melanomas of skin, dx or hx,Basal cell carcino	X:64823258:A:G	4.46E-02
rs5964750	X:64250161:T:A	1.89E-10	total	A	-0.0401561	0.00630382			NA		X:64823258:T:C	1.77E-01
rs5016561	X:54057003:T:A	2.48E-08	total	T	-0.0243345	0.00436468			0.19225937		X:64823258:T:C	9.42E-06
rs4598397	X:64872658:A:G	2.33E-11	total	G	-0.0446278	0.00667663			NA		X:64823258:T:C	8.99E-01
rs112681238	X:65392219:T:G	5.53E-11	total	G	-0.0449314	0.00685292			NA	Hypertension,Essential hypertension,Hyperten	X:64823258:T:C	8.97E-01
rs60137203	X:72490395:C:T	3.23E-09	total	T	0.0537771	0.00908437			0.59719948		X:75191364:G:T	8.01E-04

The androgen receptor is located on Chromosome X and may explain the numerous novel association on the chromosome. To test whether these association were due to incomplete conditional analysis, we conducted conditional X chromosome analysis. Three X chromosome variants were not significant after conditioning on the nearest X chromosome variant, resulting in 6 novel variants. We have included results of conditional analysis in **Supplementary Table 4**.

6) Figure 5B - there is something drastically wrong here because the estimate for 'all' is much lower than in each of the ancestry groups separately.

We have revised Figure 5 to incorporate new Cox proportional hazards analysis that better accounts for age. In this revised figure, estimates for all are fairly consistent with other ancestry groups. In cases where the estimate is much higher (Figure 5D, SHBG), the increased effect size may be due to a better powered analysis. Additionally, there are other individuals that did not cluster within the EUR, AFR, AMR, EAS included in the analysis, which may explain the higher effect size.

Figure 5: Association of Total Testosterone, SHBG GS and Hypogonadism GRS with Disease Risk in MVP. CoxPH Hazards ratio of association of total testosterone GS, SHBG GS and hypogonadism GRS with diabetes (A), hyperlipidemia (B), obesity (C), gout (D), and liver disorders (E) in all, EUR, AFR, AMR and EAS ancestry in the MVP.

Other comments:

Page 5 "Effects of Genetic Variants Influencing Male Sex Hormones are Consistent across Ancestry Groups" - this is an odd title considering the text highlights that "Variant effects are consistently different in the East Asian group" - and is inconsistent with the next title on Page 6. But is this because of the very small sample size of East Asians? Is there actual statistical heterogeneity?

We reevaluated the ancestry-specific effects of trans-ancestry variants. We found in the cases where there were different directions for beta values, the associations were not significant. Please see above comment

Page 8 - add main text citation for Fig S7

We have added a main text citation for Figure S7.

Figure 2: legend: clarify that gene implication was performed using GTEx cis-eQTL gene expression information

We have revised Figure 2 legend to clarify that gene implication was performed using GTEx information.

Figure 2: Genes Implicated in Total Testosterone, Free Testosterone, SHBG and Hypogonadism GWAS Analysis. Implication performed using GTEx cis-eQTL gene expression information. Network plot of genes (coral), variants (gray) and phenotype (teal). Scatter points are scaled by number of links.

Figure 4: title "Polygenic Score Construction and Validation for..." - clarify that this is simply "internal validation" as this figure does not include additional non-discovery data.

We have revised Figure 4 caption to clarify that this is internal validation.

Figure 4: Total Testosterone, SHBG GS and Hypogonadism GRS Construction and Internal Validation. Quantile plots of total testosterone GS (A), SHBG GS (B) and hypogonadism GRS (C) and corresponding levels of total testosterone (ng/dL), SHBG levels (nmol/dl) and odds of hypogonadism, respectively for EUR, AFR, AMR and EAS ancestry groups.

Reviewer #1 (Remarks to the Author):

The authors have responded to comments but I still feel the manuscript needs a lot of tightening up. It is worth asking someone with extensive GWAS experience to have a good read and check. I sense it was written by experts in the phenotypes rather than the genotypes. Here are some things that may help:

Thank you for this feedback. We have incorporated feedback from Dr. Tiffany Amariuta, a statistical geneticist whose lab focuses on studying complex traits in global populations. A significant suggestion that we have incorporated is to do a meta-analysis of the MVP studies rather than a total MVP population GWAS to reduce any population-specific artifacts. As a result, we have now rearranged the article to first start with ancestry specific associations followed by the meta-analysis. We have retained the majority of original associations with this more rigorous approach. We also have clarified wording and methodology in several areas of the article.

1. In the abstract its important to reword to make clear you've not found genes but variants . This is also important in the main text - rather than refer to associations with genes (intro for example) I suggest « associations of variants in or near genes »

The wording has been updated throughout to refer to variants instead of genes.

2. The wording of the Susie analysis page 4 and 5 is a little odd. What do the authors mean when they claim an association to be causal ? Do they mean that the PIP evidence is string that the lead variant is the causal variant ? If so I suggest it is better to talk about the « causal variant »

The terminology has been updated as suggested:

The associations of 49 (31.2%) lead variants had a high PIP> 0.10.

3. Likewise the wording is confusing in other places for example « Of the significant associations in the MVP, 63 total testosterone, 13 SHBG and 1 free testosterone associations were found in UK Biobank summary statistics, respectively » I think the authors mean the variant was available for look up not that the association was found (the « association was found » implies that the association between the variant and testosterone levels was replicated .

We have adjusted this analysis to use a proxy variant as suggested and have updated the methods and article to represent the total % of variants which were validated.

Of the MVP EUR GWAS variants, 87.2% (89.1% total testosterone, 100% SHBG, 33.3% free testosterone) of EUR GWAS variants were validated in the UK Biobank.

4. Of the 164 variants found to be associated with sex hormone levels in MVP, less than half were available in the Uk Biobank data which seems strange - the full GWAS stats from Ruth et al. Are available on line at the EBI GWAS catalogue and close LD proxies could be used if specific rsIDs are not available.

We have used proxies for any variants not found in UK Biobank summary statistics. Proxies used are given in Supplementary Table 3 and 6.

5. Again on lines 169-170 the wording is confusing - « associations were found » implies replication not just that the variant was present

The wording has been updated as suggested. We now use the terminology, “replicated”.

6. Lines 237 onwards - if the dementia result did not reach bonferroni significance I suggest do not mention it could raise expectations .

Thank you for this feedback. We received similar feedback from another reviewer. We have clarified in the discussion that the results were not significant, but we retained a brief explanation about why we looked for a link to begin with.

7. Lines 286 in the discussion . Is it possible the « prevalence » of hypogonadism in MVP is not a true reflection of the population prevalence but a feature of clinical testing of men suspected of having low levels ? I realise you talk about this but in places you suggest the differences with the uk Biobank could be due mainly to age .

Thank you for this feedback. We have better highlighted the selection bias within this study in the discussion by adding the following text: "However, the difference in hypogonadism prevalence is likely inflated due to a selection bias. While all participants within the UK Biobank received testosterone testing upon enrollment, the participants of MVP only received endocrine testing when ordered within a clinical setting, likely resulting in a cohort where many participants had clinical suspicion of low testosterone levels."

Reviewer #2 (Remarks to the Author):

The paper is overall much clearer and specifically the authors have significantly improved the level of detail provided in the text to support their stated results. I have 2 further comments.

1. The more extensive PGS-to-disease analysis now provides much greater substantiation for the claimed "multitude of roles for testosterone in human disease" as the PGS provides much stronger evidence for a causal effect of testosterone than the PheWAS which tests individual SNPs. While they took a logical approach to choose the disease outcomes for PGS testing ("We focused on top 50 significant associations from PheWAS analysis which included diabetes, hyperlipidemia, gout, liver disease, and obesity"), unfortunately this excludes some outcomes of high clinical interest and which were highlighted by missense variant associations. The most important of these is prostate cancer (or PSA as a proxy) which should be added to the PGS analysis and also possibly osteoporosis. While the paper finds overall possible benefits for testosterone on metabolic health outcomes, prostate cancer is the main potential adverse effect.

We agree that exploration of prostate cancer and testosterone is valuable. As a result, we have expanded our PheWAS analysis to extend beyond metabolic conditions. Several variants were also linked to cardiac conditions and we have now included significant associations in the article. Since none of the variant were associated with osteoporosis on PheWAS analysis, we did not include the outcome in GS and GRS analyses.

Figure 5: Association of Total Testosterone, SHBG GS and Hypogonadism GRS with Disease Risk in MVP. CoxPH Hazards ratio of association of total testosterone GS, SHBG GS and hypogonadism GRS with diabetes (A), hyperlipdemia (B), obesity (C), gout (D), liver disorders (E) and cardiac disorders (F) in EUR, AFR, AMR and EAS ancestry in the MVP.

Figure S9: Association of Total Testosterone, SHBG GS and Hypogonadism GRS with Testicular Dysfunction, Prostate Cancer, Dementia Risk. CoxPH Hazards ratio of total testosterone GS, SHBG GS and hypogonadism GRS and phecode 571 - chronic liver disease (A), 572 - ascites (B), and 573 - liver necrosis (B) in EUR, AFR, AMR, EAS ancestry groups in MVP.

Figure S10: Association of Total Testosterone, SHBG GS and Hypogonadism GRS with Testicular Dysfunction, Prostate Cancer, Dementia Risk. CoxPH Hazards ratio of total testosterone GS, SHBG GS and hypogonadism GRS and phecode 257 - testicular dysfunction (A), phecode 185 - prostate cancer (B), and phecode 290 - dementia (B) in EUR, AFR, AMR, EAS ancestry groups in MVP.

2. Testosterone and dementia. In response to my previous comment, the authors have responded that these associations were not significant after multiple testing (Lines 238-240). The relevant discussion should therefore be appropriately guarded (279-285) as it currently implies that a clear result was found.

Thank you for this feedback. We have updated the discussion to explicitly state that the associations were not significant.

REVIEWERS' COMMENTS

Reviewer #1 (Remarks to the Author):

The authors have tightened up the manuscript. I have some final very small comments:

1. Table S3 is in an incomprehensible pdf format. I strongly suggest to reformat all tables into spreadsheets so that readers can easily scan all the information, if this is not a journal requirement I suggest this will help readability and citability !

All of the tables are now in Microsoft Excel spreadsheet format to improve readability.

2. Almost all of the associations identified were found in the UKBiobank, and the authors claim "and four total testosterone and two free testosterone testosterone 139 associations were novel" but I cannot see any information about these 6 variants in the tables. These novel findings should be highlighted in a table, but also the UKB data for these 6 needs to be shown, could they be false positives ?

We have now included Table S4, to highlight the 6 variants which were not validated in the UK Biobank. One variant was located on chromosome X, 2 on chromosome 14, 2 on chromosome 1 and lastly 1 on chromosome 6. Generally, the effect sizes and minor allele frequencies were consistent, suggesting similar behavior of variants between cohorts, despite differences in p-values.

There are a number of possibilities for why these MVP variants may not have been validated in the UK Biobank:

- Population differences in MVP versus UK Biobank
- Association with a variable linked to total and free testosterone levels not accounted for by the covariates included in GWAS analysis

Variants may be false positives; however, we used strict GWAS significance threshold ($5e-08$) for identifying significant variants. QQ and Manhattan plots do not suggest genomic inflation. We do agree that "novel" is probably inappropriate wording and have instead included analyses about minor allele frequencies and effect sizes. Further investigation is needed to determine if these are novel variants.

4. line 258 discussion . The authors claim to have "discovered" associations between testosterone gene variants and diseases but some of these were described in Ruth et al. ?

We revised the wording to state that we "identified" the associations. Later in the paragraph we state that these associations were consistent with Ruth et al. findings:

Using the GS for total testosterone and SHBG and the hypogonadism genetic risk score (GRS) we developed from our GWAS analyses, we identified associations between testosterone gene variants and the risk of certain diseases, especially metabolic diseases such as diabetes, gout, and hyperlipidemia. Low testosterone levels are associated with increased insulin resistance in men with both type 1 and type 2 diabetes, and insulin sensitivity and testosterone levels are hypothesized to influence one another. The association between testosterone GS and diabetes risk identified in this study is consistent with the findings in Ruth et al., in which a Mendelian randomization analysis suggested that higher testosterone reduced the risk of type 2 diabetes in men.

Reviewer #2 (Remarks to the Author):

Lines 194-197 & Figure 4 - specify 'decile' rather than 'quantile' which too vague. Also, the text wrongly describes the 'bottom quantile' as the comparator - in Figure 4 it appears to be the 6th decile.

We have replaced quantile with decile in Figure 4, Figure S7 and lines 194-197.

In Figure 4, the comparator is the 6th decile when determining decile odds for each group. However, in the text, the comparison between the top and bottom deciles demonstrates stratification of odds of hypogonadism with the constructed PRS.

Lines 203-208 & Figure S8 - for Mendelian randomisation associations, clarify which exposure set was used, i.e. which ancestry or was it from the multi-ancestry meta-analysis? (in the preceding para ancestry specific GRS are used).

We only used European individuals for the exposure set. We have revised Figure S8 and lines 203-208 to clarify.

Cox proportional hazards analyses - clearly describe what is the multiple test significance threshold used here? The authors previously stated that associations with dementia did not reach this threshold. But now they describe an association with chronic liver disease at " $p < .04$ ".

We used Benjamini-Hochberg multiple test correction for CoxPH analyses. The AFR association with chronic liver disease is not significant after multiple test correction and we revised the statement to reflect the current analysis.

Supplementary Table 10 also includes all the results from the CoxPH analyses.

The lack of association with prostate cancer is surprising. Are numbers of cases of disease in the analyzed MVP sample shown (I could not find it)? Might these be insufficient?

We have included the number of cases and controls in each CoxPH analysis in Supplementary Table 10. At least 1000 cases were available for CoxPH analysis in all ancestry groups except EAS.